# CONFORMAL RISK CONTROL

**Anastasios N. Angelopoulos[1], Stephen Bates[2], Adam Fisch[2], Lihua Lei[3], Tal Schuster[4]**

[1]UC Berkeley    [2]MIT    [3]Stanford    [4]Google Research

## ABSTRACT

We extend conformal prediction to control the expected value of any monotone loss function. The algorithm generalizes split conformal prediction together with its coverage guarantee. Like conformal prediction, the conformal risk control procedure is tight up to an $\mathcal{O}(1/n)$ factor. We also introduce extensions of the idea to distribution shift, quantile risk control, multiple and adversarial risk control, and expectations of U-statistics. Worked examples from computer vision and natural language processing demonstrate the usage of our algorithm to bound the false negative rate, graph distance, and token-level F1-score.

## 1 INTRODUCTION

We seek to endow some pre-trained machine learning model with guarantees on its performance as to ensure its safe deployment. Suppose we have a base model $f$ that is a function mapping inputs $x \in \mathcal{X}$ to values in some other space, such as a probability distribution over classes. Our job is to design a procedure that post-processes the output of $f$ to give it a statistical safety property.

Split conformal prediction (Vovk et al., 2005; Papadopoulos et al., 2002), which we will refer to simply as "conformal prediction", has been useful in areas such as computer vision (Angelopoulos et al., 2021b) and natural language processing (Fisch et al., 2021) to provide such a guarantee. By measuring the model's performance on a *calibration dataset* $\{(X_i, Y_i)\}_{i=1}^n$ of feature-response pairs, conformal prediction post-processes the model to construct prediction sets that bound the *miscoverage*,

$$\mathbb{P}\Big(Y_{n+1} \notin \mathcal{C}(X_{n+1})\Big) \leq \alpha, \tag{1}$$

where $(X_{n+1}, Y_{n+1})$ is a new test point, $\alpha$ is a user-specified error rate (e.g., 10%), and $\mathcal{C}$ is a function of the model and calibration data that outputs a prediction set. Note that $\mathcal{C}$ is formed using the first $n$ data points, and the probability in (1) is over the randomness in all $n + 1$ data points (i.e., the draw of both the calibration points $1, \dots, n$ and the test point $n + 1$).

In this work, we extend conformal prediction to prediction tasks where the natural notion of error is not simply miscoverage. In particular, our main result is that a generalization of conformal prediction provides guarantees of the form

$$\mathbb{E}\Big[\ell\big(\mathcal{C}_\lambda(X_{n+1}), Y_{n+1}\big)\Big] \leq \alpha, \tag{2}$$

for any bounded *loss function* $\ell$ that shrinks as $\mathcal{C}_\lambda(X_{n+1})$ grows, and $\lambda$ is an input parameter that controls the growth of $\mathcal{C}_\lambda(X_{n+1})$. We call this *conformal risk control*. Note that (2) recovers the conformal miscoverage guarantee in (1) when using the miscoverage loss, $\ell\big(\mathcal{C}_\lambda(X_{n+1}), Y_{n+1}\big) = \mathbb{1}\{Y_{n+1} \notin \mathcal{C}_\lambda(X_{n+1})\}$. However, our algorithm also extends conformal prediction to situations where other loss functions, such as the false negative rate (FNR), are more appropriate.

As an example, consider multilabel classification, where the $Y_i \subseteq \{1, ..., K\}$ are sets comprising a subset of $K$ classes. Creating sets that contain *all* the classes may be too conservative if $K$ is massive; instead, given a trained multilabel classifier $f : \mathcal{X} \to [0, 1]^K$, we want to output sets that include a large fraction of the true classes in $Y_i$. To that end, we post-process the model's raw outputs into the set of classes with sufficiently high scores, $\mathcal{C}_\lambda(x) = \{k : f(X)_k \geq 1 - \lambda\}$, where the main parameter of the algorithm $\lambda \in [0, 1]$ is a threshold. Note that as the threshold $\lambda$ grows, we include more classes in $\mathcal{C}_\lambda(x)$—i.e., it becomes more conservative. In this case, conformal risk control finds a threshold value $\hat{\lambda}$ that controls the fraction of missed classes, i.e., the expected value

of $\ell\big(\mathcal{C}_{\hat{\lambda}}(X_{n+1}), Y_{n+1}\big) = 1 - |Y_{n+1} \cap \mathcal{C}_\lambda(X_{n+1})|/|Y_{n+1}|$. Setting $\alpha = 0.1$ would ensure that our algorithm produces sets $\mathcal{C}_{\hat{\lambda}}(X_{n+1})$ containing $\geq 90\%$ of the true classes in $Y_{n+1}$ on average.

## 1.1 ALGORITHM AND PREVIEW OF MAIN RESULTS

Formally, we will consider post-processing the predictions of the model $f$ to create a function $\mathcal{C}_\lambda(\cdot)$. The function has a parameter $\lambda$ that encodes its conservativeness: larger $\lambda$ values yield more conservative outputs (e.g., larger prediction sets). To measure the quality of $\mathcal{C}_\lambda$, we consider a loss function $\ell(\mathcal{C}_\lambda(x), y) \in (-\infty, B]$ for some $B < \infty$.[1] We require this loss to be non-increasing in $\lambda$. Our goal is to choose $\hat{\lambda}$ based on the observed data $\{(X_i, Y_i)\}_{i=1}^n$ so that risk control as in (2) holds.

We now rewrite this same task in a more notationally convenient and abstract form. Consider an exchangeable collection of non-increasing, bounded, random functions $L_i : \Lambda \to (-\infty, B]$, $i = 1, \ldots, n+1$, where $\Lambda$ is the space of all inputs (e.g., 'thresholds') to the function $L_i(\lambda)$. Throughout the paper, we assume $\lambda_{\max} := \sup \Lambda \in \Lambda$, so that $L_i(\lambda_{\max})$ is well-defined and satisfies $L_i(\lambda_{\max}) \leq \alpha$ for any $\alpha$ by design (i.e., $\alpha$ is achievable). We seek to use the first $n$ functions to choose a value of the parameter, $\hat{\lambda}$, so that the risk on the unseen function is controlled:

$$\mathbb{E}\left[L_{n+1}(\hat{\lambda})\right] \leq \alpha. \tag{3}$$

We are primarily motivated by the case where $L_i(\lambda) = \ell(\mathcal{C}_\lambda(X_i), Y_i)$, in which case the guarantee in (3) coincides with risk control as in (2).

Now we describe the algorithm. Let $\widehat{R}_n(\lambda) = (L_1(\lambda) + \ldots + L_n(\lambda))/n$. Given any desired risk level upper bound $\alpha \in (-\infty, B)$, define

$$\hat{\lambda} = \inf\left\{\lambda : \frac{n}{n+1}\widehat{R}_n(\lambda) + \frac{B}{n+1} \leq \alpha\right\} = \inf\left\{\lambda : \widehat{R}_n(\lambda) \leq \alpha - \frac{B-\alpha}{n}\right\}. \tag{4}$$

Since $\widehat{R}_n(\lambda)$ is monotone, we can efficiently search for $\hat{\lambda}$ using binary search to arbitrary precision. When the set is empty, we define $\hat{\lambda} = \lambda_{\max}$. Our proposed *conformal risk control* algorithm is to deploy $\hat{\lambda}$ on the forthcoming test point. Our main result is that this algorithm satisfies (3). Intuitively, we can see that this algorithm reduces to searching for a value of $\lambda$ that results in a slightly conservative empirical risk—that gets less conservative when the difference between the worst-case risk ($B$) and the desired risk ($\alpha$) is smaller, or the calibration set size ($n$) is larger.

Moreover, when the $L_i$ are i.i.d. from a continuous distribution, we can show that the algorithm satisfies a tight lower bound saying it is not too conservative,

$$\mathbb{E}\left[L_{n+1}(\hat{\lambda})\right] \geq \alpha - \frac{2B}{n+1}.$$

We show the reduction from conformal risk control to conformal prediction in Appendix A. Furthermore, if the risk is non-monotone, then this algorithm does not control the risk; we discuss this in Section 2.3. Finally, we provide both practical examples using real-world data and several theoretical extensions of our procedure in Sections 3 and 4, respectively.

## 1.2 RELATED WORK

Conformal prediction was developed by Vladimir Vovk and collaborators beginning in the late 1990s (Vovk et al., 1999; 2005), and has recently become a popular uncertainty estimation tool in the machine learning community, due to its favorable model-agnostic, distribution-free, finite-sample guarantees. See Angelopoulos & Bates (2021) for a modern introduction to the area or Shafer & Vovk (2008) for a more classical alternative. As previously discussed, in this paper we primarily build on *split conformal prediction* (Papadopoulos et al., 2002); statistical properties of this algorithm including the coverage upper bound were studied in Lei et al. (2018). Recently there have been many extensions of the conformal algorithm, mainly targeting deviations from exchangeability (Tibshirani et al., 2019; Gibbs & Candes, 2021; Barber et al., 2022; Fannjiang et al., 2022) and

---

[1]Note that any unbounded loss can be transformed to a bounded loss. For example, any unbounded loss $\ell(\lambda)$ on the positive reals can be transformed by taking the inverse tangent, i.e., $\ell'(\lambda) = \arctan(\ell(\lambda))$. As long as the transformation is monotone, a loss of $\alpha$ on the original loss corresponds exactly to a loss of $\alpha'$ on the new loss; controlling this transformed risk may be enough in practice.

improved conditional coverage (Barber et al., 2020; Romano et al., 2019; Guan, 2020; Romano et al., 2020; Angelopoulos et al., 2021b). Most relevant to us is recent work on risk control in high probability (Vovk, 2012; Bates et al., 2021; Angelopoulos et al., 2021a) and its applications (Park et al., 2020; Fisch et al., 2022; Schuster et al., 2021; 2022; Sankaranarayanan et al., 2022; Angelopoulos et al., 2022a;b, *inter alia*). However, while Bates et al. (2021) and Angelopoulos et al. (2021a) operate in similar mathematical settings and we reuse much of their notation, the algorithm presented herein differs greatly. It is far more sample-efficient, simpler, and provides a guarantee in expectation versus a guarantee in probability. Our algorithm is entirely different—no existing algorithm gives guarantees in expectation for risk control—and its validity proof is mathematically unrelated to these previous works. See Appendix B for a detailed discussion and comparison of these algorithms, including experiments.

To further elaborate on the difference between our work and the broader existing literature, first consider conformal prediction. The purpose of conformal prediction is to provide coverage guarantees of the form in (1). The guarantee available through conformal risk control, (3), strictly subsumes that of conformal prediction; it is generally impossible to recast risk control as coverage control. As a second question, one might ask whether (3) can be achieved through standard statistical machinery, such as uniform concentration inequalities. Though it is possible to integrate a uniform concentration inequality to get a bound in expectation, this strategy tends to be excessively loose both in theory and in practice (see, e.g., the bound of Anthony & Shawe-Taylor (1993)). The technique herein avoids these complications; it is simpler than concentration-based approaches, practical to implement, and tight up to a factor of $1/n$, which is comparatively faster than concentration would allow.

## 2 THEORY

In this section, we establish the core theoretical properties of conformal risk control. All proofs, unless otherwise specified, are deferred to Appendix E.

### 2.1 RISK CONTROL

We first show that the proposed algorithm leads to risk control when the loss is monotone.

**Theorem 1.** *Consider a sequence of exchangeable random loss functions, $\{L_i(\lambda)\}_{i=1}^{n+1}$, where $L_i : \Lambda \to \mathbb{R}$ which are non-increasing in $\lambda$, right-continuous, and for $\lambda_{\max} = \sup \Lambda \in \Lambda$, satisfy*

$$L_i(\lambda_{\max}) \le \alpha, \quad \sup_\lambda L_i(\lambda) \le B < \infty \text{ almost surely.} \tag{5}$$

*Then*

$$\mathbb{E}[L_{n+1}(\hat{\lambda})] \le \alpha.$$

*Proof.* Let $\widehat{R}_{n+1}(\lambda) = (L_1(\lambda) + \ldots + L_{n+1}(\lambda))/(n+1)$ and

$$\hat{\lambda}' = \inf \left\{ \lambda \in \Lambda : \widehat{R}_{n+1}(\lambda) \le \alpha \right\}.$$

Since $\inf_\lambda L_i(\lambda) = L_i(\lambda_{\max}) \le \alpha$, $\hat{\lambda}'$ is well-defined almost surely. Since $L_{n+1}(\lambda) \le B$, we know $\widehat{R}_{n+1}(\lambda) = \frac{n}{n+1}\widehat{R}_n(\lambda) + \frac{L_{n+1}(\lambda)}{n+1} \le \frac{n}{n+1}\widehat{R}_n(\lambda) + \frac{B}{n+1}$. Thus,

$$\frac{n}{n+1}\widehat{R}_n(\lambda) + \frac{B}{n+1} \le \alpha \implies \widehat{R}_{n+1}(\lambda) \le \alpha.$$

This implies $\hat{\lambda}' \le \hat{\lambda}$ when the LHS holds for some $\lambda \in \Lambda$. When the LHS is above $\alpha$ for all $\lambda \in \Lambda$, by definition, $\hat{\lambda} = \lambda_{\max} \ge \hat{\lambda}'$. Thus, $\hat{\lambda}' \le \hat{\lambda}$ almost surely. Since $L_i(\lambda)$ is non-increasing in $\lambda$,

$$\mathbb{E}\left[L_{n+1}(\hat{\lambda})\right] \le \mathbb{E}\left[L_{n+1}(\hat{\lambda}')\right]. \tag{6}$$

Let $E$ be the multiset of loss functions $\{L_1, \ldots, L_{n+1}\}$. Then $\hat{\lambda}'$ is a function of $E$, or, equivalently, $\hat{\lambda}'$ is a constant conditional on $E$. Additionally, $L_{n+1}(\lambda)|E \sim \text{Uniform}(\{L_1, ..., L_{n+1}\})$ by exchangeability. These facts combined with the right-continuity of $L_i$ imply

$$\mathbb{E}\left[L_{n+1}(\hat{\lambda}') \mid E\right] = \frac{1}{n+1}\sum_{i=1}^{n+1} L_i(\hat{\lambda}') \le \alpha.$$

The proof is completed by the law of total expectation and (6). $\qquad\square$

## 2.2 A TIGHT RISK LOWER BOUND

Next we show that the conformal risk control procedure is tight up to a factor $2B/(n+1)$ that cannot be improved in general. The proof will rely on a form of continuity that generalizes the assumption of continuous non-conformity scores used for the standard conformal proof. Define the *jump function*, which quantifies the size of the discontinuity in a right-continuous input function $l$ at point $\lambda$, a $J(l, \lambda) = \lim_{\epsilon \to 0^+} l(\lambda - \epsilon) - l(\lambda)$. If the probability that $L_i$ has a discontinuity at any pre-specified $\lambda$ is $\mathbb{P}(J(L_i, \lambda) > 0) = 0$, then the conformal risk control procedure is not too conservative.

**Theorem 2.** *In the setting of Theorem 1, further assume that the $L_i$ are i.i.d., $L_i \geq 0$, and for any $\lambda$, $\mathbb{P}\left(J(L_i, \lambda) > 0\right) = 0$. Then*

$$\mathbb{E}\Big[L_{n+1}\big(\hat{\lambda}\big)\Big] \geq \alpha - \frac{2B}{n+1}.$$

This bound is tight for general monotone loss functions, as we show next.

**Proposition 1.** *In the setting of Theorem 2, for any $\epsilon > 0$, there exists a loss function and $\alpha \in (0, 1)$ such that*

$$\mathbb{E}\left[L_{n+1}\big(\hat{\lambda}\big)\right] \leq \alpha - \frac{2B - \epsilon}{n+1}.$$

Since we can take $\epsilon$ arbitrarily close to zero, the factor $2B/(n+1)$ in Theorem 2 is required. Conformal prediction—both the algorithm and the guarantee—is *exactly equivalent* to conformal risk control when the loss function is an indicator, including the tighter lower bound (see Appendix A).

## 2.3 CONTROLLING GENERAL LOSS FUNCTIONS

We next show that the conformal risk control algorithm does *not* control the risk if the $L_i$ are not assumed to be monotone. In particular, (3) does not hold. We show this by example.

**Proposition 2.** *For any $\epsilon$, there exists a non-monotone loss function such that*

$$\mathbb{E}\left[L_{n+1}\big(\hat{\lambda}\big)\right] \geq B - \epsilon.$$

Notice that for any desired level $\alpha$, the expectation in (3) can be arbitrarily close to $B$. Since the function values here are in $[0, B]$, this means that even for bounded random variables, risk control can be violated by an arbitrary amount—unless further assumptions are placed on the $L_i$. However, the algorithms developed may still be appropriate for near-monotone loss functions. Simply 'monotonizing' all loss functions $L_i$ and running conformal risk control will guarantee (3), but this strategy will only be powerful (i.e., not conservative) if the loss is near-monotone. For concreteness, we describe this procedure below as a corollary of Theorem 1.

**Corollary 1.** *Allow $L_i(\lambda)$ to be any (possibly non-monotone) function of $\lambda$ satisfying 5. Take*

$$\tilde{L}_i(\lambda) = \sup_{\lambda' \geq \lambda} L_i(\lambda'), \;\; \tilde{R}_n(\lambda) = \frac{1}{n}\sum_{i=1}^{n} \tilde{L}_i(\lambda) \;\; and \;\; \tilde{\lambda} = \inf\left\{\lambda : \frac{n}{n+1}\tilde{R}_n(\lambda) + \frac{B}{n+1} \leq \alpha\right\}.$$

*Then,*

$$\mathbb{E}\left[L_{n+1}(\tilde{\lambda})\right] \leq \alpha.$$

If the loss function is already monotone, then $\tilde{\lambda}$ reduces to $\hat{\lambda}$. We propose a further algorithm for picking $\lambda$ in Appendix C that provides an asymptotic risk-control guarantee for *non-monotone* loss functions. However, this algorithm again is only powerful when the risk $\mathbb{E}[L_{n+1}(\lambda)]$ is near-monotone and reduces to the standard conformal risk control algorithm when the loss is monotone.

## 3 EXAMPLES

To demonstrate the flexibility and empirical effectiveness of the proposed algorithm, we apply it to four tasks across computer vision and natural language processing. All four loss functions are non-binary, monotone losses bounded by $1$. They are commonly used within their respective application domains. Our results validate that the procedure bounds the risk as desired and gives useful outputs

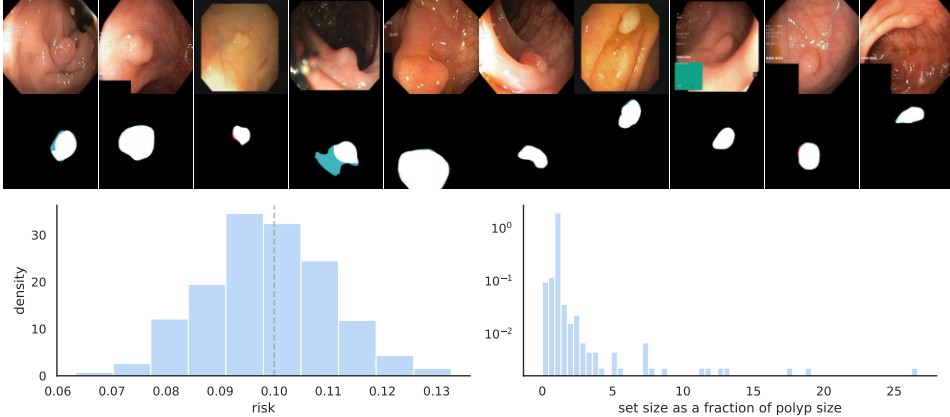

Figure 1: **FNR control in tumor segmentation**. The top figure shows examples of our procedure with correct pixels in white, false positives in blue, and false negatives in red. The bottom plots report FNR and set size over 1000 independent random data splits. The dashed gray line marks $\alpha$.

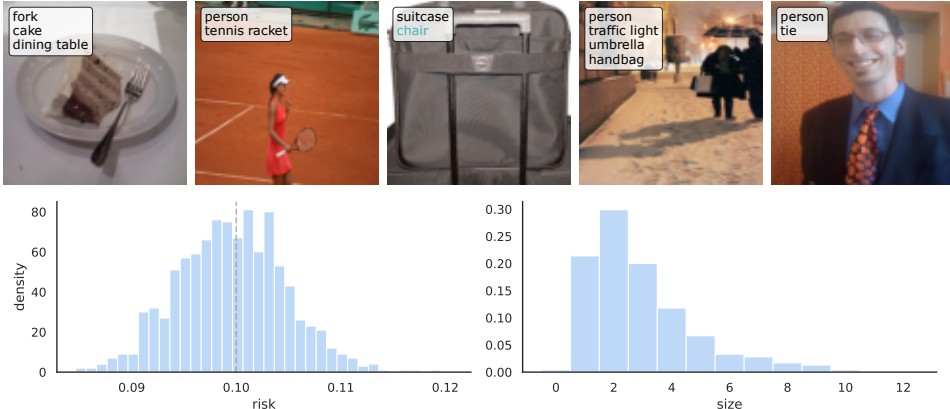

Figure 2: **FNR control on MS COCO**. The top figure shows examples of our procedure with correct classes in black, false positives in blue, and false negatives in red. The bottom plots report FNR and set size over 1000 independent random data splits. The dashed gray line marks $\alpha$.

to the end-user. We note that the choices of $\mathcal{C}_\lambda$ used herein are *only for the purposes of illustration*; any nested family of sets will work. For each example use case, for a representative $\alpha$ (details provided for each task) we provide both qualitative results and quantitative histograms of the risk and set sizes over 1000 random data splits that demonstrate valid risk control (i.e., with mean $\leq \alpha$).

## 3.1 FNR CONTROL IN TUMOR SEGMENTATION

In tumor segmentation, our input is a $d \times d$ image and our label is a set of pixels $Y_i \in \wp\left(\{(1,1),(1,2),...,(d,d)\}\right)$, with $\wp$ the power set. We use an image segmentation model $f : \mathcal{X} \to [0,1]^{d \times d}$ outputting a probability for each pixel and measure loss as the fraction of false negatives,

$$L_i^{\mathrm{FNR}}(\lambda) = 1 - \frac{|Y_i \cap \mathcal{C}_\lambda(X_i)|}{|Y_i|}, \text{ where } \mathcal{C}_\lambda(X_i) = \{y : f(X_i)_y \geq 1 - \lambda\}. \tag{7}$$

The expected value of $L_i^{\mathrm{FNR}}$ is the FNR. Since $L_i^{\mathrm{FNR}}$ is monotone, so is the FNR. Thus, we use the technique in Section 2.1 to pick $\hat{\lambda}$ by (4) that controls the FNR on a new point, which guarantees:

$$\mathbb{E}\left[L_{n+1}^{\mathrm{FNR}}(\hat{\lambda})\right] \leq \alpha. \tag{8}$$

For evaluating the proposed procedure we pool data from several online open-source gut polyp segmentation datasets: Kvasir, Hyper-Kvasir, CVC-ColonDB, CVC-ClinicDB, and ETIS-Larib. We choose a PraNet (Fan et al., 2020) as our base model $f$ and used $n = 1000$, and evaluated risk control with the 781 remaining validation data points. We report results with $\alpha = 0.1$ in Figure 1. The mean and standard deviation of the risk over 1000 trials are 0.0987 and 0.0114, respectively.

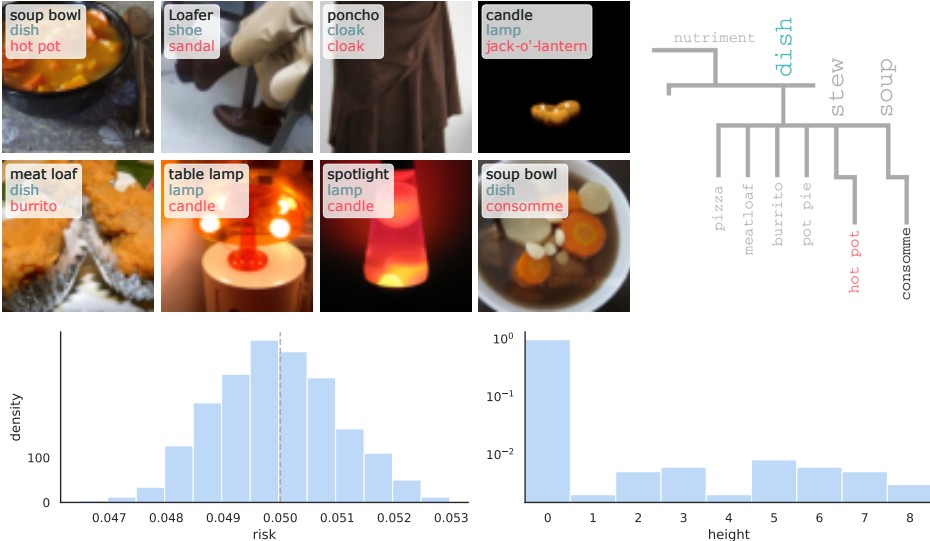

Figure 3: **Control of graph distance on hierarchical ImageNet**. The top figure shows examples of our procedure with correct classes in black, false positives in blue, and false negatives in red. The bottom plots report our minimum hierarchical distance loss and set size over 1000 independent random data splits. The dashed gray line marks $\alpha$.

### 3.2 FNR CONTROL IN MULTILABEL CLASSIFICATION

In the multilabel classification setting, our input $X_i$ is an image and our label is a set of classes $Y_i \subset \{1, \ldots, K\}$ for some number of classes $K$. Using a multiclass classification model $f : \mathcal{X} \to [0, 1]^K$, we form prediction sets and calculate the number of false positives exactly as in (7). By Theorem 1, picking $\hat{\lambda}$ as in (4) again yields the FNR-control guarantee in (8). We evaluate on the Microsoft Common Objects in Context (MS COCO) dataset (Lin et al., 2014), a large-scale 80-class multiclass classification task commonly used in computer vision. We choose a TResNet (Ridnik et al., 2020) as our model $f$ and used $n = 4000$, and evaluated risk control with 1000 validation data points. We report results with $\alpha = 0.1$ in Figure 2. The mean and standard deviation of the risk over 1000 trials are 0.0996 and 0.0052, respectively. The results indicate that the risk is almost exactly controlled, the spread is not too wide, and the set sizes are reasonable, not overly inflated.

### 3.3 CONTROL OF GRAPH DISTANCE IN HIERARCHICAL IMAGE CLASSIFICATION

In the $K$-class hierarchical classification setting, our input $X_i$ is an image and our label is a leaf node $Y_i \in \{1, ..., K\}$ on a tree with nodes $\mathcal{V}$ and edges $\mathcal{E}$. Using a single-class classification model $f : \mathcal{X} \to \Delta^K$, we calibrate a loss in graph distance between the interior node we select and the closest ancestor of the true class. For any $x \in \mathcal{X}$, let $\hat{y}(x) = \arg\max_k f(x)_k$ be the class with the highest estimated probability. Further, let $d : \mathcal{V} \times \mathcal{V} \to \mathbb{Z}$ be the function that returns the length of the shortest path between two nodes, let $\mathcal{A} : \mathcal{V} \to 2^\mathcal{V}$ be the function that returns the ancestors of its argument, and let $\mathcal{P} : \mathcal{V} \to 2^\mathcal{V}$ be the function that returns the set of leaf nodes that are descendants of its argument. We also let $g(v, x) = \sum_{k \in \mathcal{P}(v)} f(x)_k$ be the sum of scores of leaves descended from $v$. Further, define a hierarchical distance

$$d_H(v, u) = \inf_{a \in \mathcal{A}(v)} \{d(a, u)\}.$$

For a set of nodes $\mathcal{C}_\lambda \in 2^\mathcal{V}$, we then define the set-valued loss

$$L_i^{\text{Graph}}(\lambda) = \inf_{s \in \mathcal{C}_\lambda(X_i)} \{d_H(y, s)\}/D, \text{ where } \mathcal{C}_\lambda(x) = \bigcap_{\{a \in \mathcal{A}(\hat{y}(x)) \, : \, g(a, x) \geq -\lambda\}} \mathcal{P}(a).$$

This loss returns zero if $y$ is a child of any element in $\mathcal{C}_\lambda$, and otherwise returns the minimum distance between any element of $\mathcal{C}_\lambda$ and any ancestor of $y$, scaled by the depth $D$. Thus, it is a monotone loss function and can be controlled by choosing $\hat{\lambda}$ as in (4) to achieve the guarantee

$$\mathbb{E}\left[L_{n+1}^{\text{Graph}}(\hat{\lambda})\right] \leq \alpha.$$

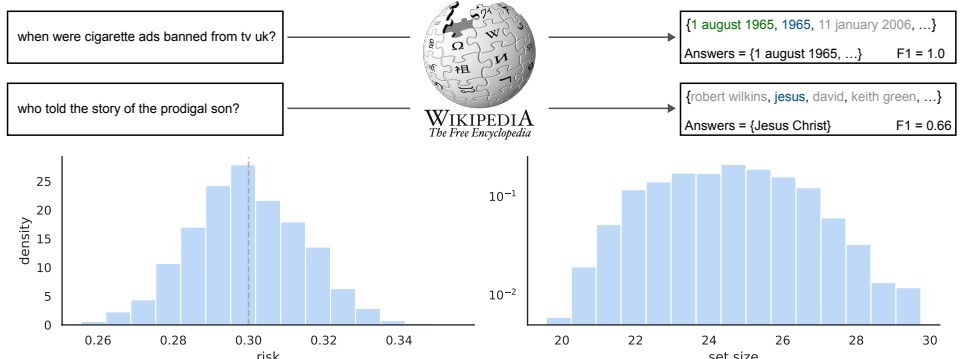

Figure 4: **F1-score control on Natural Questions**. The top figure shows examples of our procedure with fully correct answers in green, partially correct answers in blue, and false positives in gray. Note that answers that are technically correct may still be down-graded if they do not match the reference. We treat this as part of the randomness in the task. The bottom plots report the F1 risk and average set size over 1000 independent random data splits. The dashed gray line marks $\alpha$.

We use the ImageNet dataset (Deng et al., 2009), which comes with an existing label hierarchy, WordNet, of maximum depth $D = 14$. We choose a ResNet152 (He et al., 2016) for $f$ and $n = 30000$, and evaluate risk with the remaining 20000. We report results with $\alpha = 0.05$ in Figure 3. The mean and standard deviation of the risk over 1000 trials are 0.0499 and 0.0011, respectively. The results indicate that the risk is almost exactly controlled, and that the adaptively chosen resolution of the prediction appropriately encodes the model uncertainty (it is almost always a leaf node).

## 3.4 F1-SCORE CONTROL IN OPEN-DOMAIN QUESTION ANSWERING

In the open-domain question answering setting, our input $X_i$ is a question and our label $Y_i$ is a set of (possibly non-unique) correct answers. For example, the input

$$X_{n+1} = \text{"Where was Barack Obama Born?"}$$

could have the answer set

$$Y_{n+1} = \{\text{"Hawaii", "Honolulu, Hawaii", "Kapo'olani Medical Center"}\}$$

Formally, here we treat all questions and answers as being composed of sequences (up to size $m$) of tokens in a vocabulary $\mathcal{V}$—i.e., assuming $k$ valid answers, we have $X_i \in \mathcal{Z}$ and $Y_i \in \mathcal{Z}^k$, where $\mathcal{Z} := \mathcal{V}^m$. Using an open-domain question answering model that individually scores candidate output answers $f \colon \mathcal{Z} \times \mathcal{Z} \to \mathbb{R}$, we calibrate the *best* token-based F1-score of the prediction set, taken over all pairs of predictions and answers:

$$L_i^{\mathrm{F1}}(\lambda) = 1 - \max\left\{\mathrm{F1}(a, c) \colon c \in \mathcal{C}_\lambda(X_i), a \in Y_i\right\}, \text{ where } \mathcal{C}_\lambda(X_i) = \{y \in \mathcal{V}^m : f(X_i, y) \geq \lambda\}.$$

We define the F1-score following popular QA evaluation metrics (Rajpurkar et al., 2016), where we treat predictions and ground truth answers as bags of tokens and compute the geometric average of their precision and recall (while ignoring punctuation and articles {"a", "an", "the"}). Since $L_i^{\mathrm{F1}}$, as defined in this way, is monotone and upper bounded by 1, it can be controlled by choosing $\hat{\lambda}$ as in Section 2.1 to achieve the following guarantee:

$$\mathbb{E}\left[L_{n+1}^{\mathrm{F1}}(\hat{\lambda})\right] \leq \alpha.$$

We use the Natural Questions (NQ) dataset (Kwiatkowski et al., 2019), a popular open-domain question answering baseline, to evaluate our method. We use the splits distributed as part of the Dense Passage Retrieval (DPR) package (Karpukhin et al., 2020). Our base model is the DPR Retriever-Reader model (Karpukhin et al., 2020), which retrieves passages from Wikipedia that might contain the answer to the given query, and then uses a reader model to extract text sub-spans from the retrieved passages that serve as candidate answers. Instead of enumerating all possible answers to a given question, we retrieve the top several hundred candidate answers, extracted from the top 100 passages. We use $n = 2500$ calibration points, and evaluate risk control with the remaining 1110. We use $\alpha = 0.3$ (chosen empirically as the lowest F1 score which reliably results in approximately correct answers by manual validation) in Figure 4. The mean and standard deviation of the risk over 1000 trials are 0.2996 and 0.0150, respectively. The results indicate that the risk is almost exactly controlled, and that the sets are reasonably sized, scaling appropriately with question difficulty.

## 4 EXTENSIONS

We now discuss several extensions of conformal risk control to different settings and risks.

### 4.1 RISK CONTROL UNDER DISTRIBUTIONAL SHIFT

Under a distribution shift, the goal in (3) can be redefined as

$$\mathbb{E}_{(X_1,Y_1),\ldots,(X_n,Y_n)\sim P_{\text{train}},\ (X_{n+1},Y_{n+1})\sim P_{\text{test}}}\left[L_{n+1}(\hat{\lambda})\right] \leq \alpha. \tag{9}$$

Assuming that $P_{\text{test}}$ is absolutely continuous with respect to $P_{\text{train}}$ and defining $w(x,y) = \frac{dP_{\text{test}}(x,y)}{dP_{\text{train}}(x,y)}$, the weighted objective (9) can be rewritten as

$$\mathbb{E}_{(X_1,Y_1),\ldots,(X_{n+1},Y_{n+1})\sim P_{\text{train}}}\left[w(X_{n+1},Y_{n+1})L_{n+1}(\hat{\lambda})\right] \leq \alpha. \tag{10}$$

When $w$ is known and bounded, we can apply our procedure on the loss function $\tilde{L}_{n+1}(\lambda) = w(X_{n+1},Y_{n+1})L_{n+1}(\lambda)$, which is non-decreasing, bounded, and right-continuous in $\lambda$ whenever $L_{n+1}$ is. Thus, Theorem 1 guarantees that the resulting $\hat{\lambda}$ satisfies (10). For example, in the covariate shift setting, $w(X_{n+1},Y_{n+1}) = w(X_{n+1}) \triangleq \frac{dP_{\text{test}}(X_{n+1})}{dP_{\text{train}}(X_{n+1})}$. In this case, we can achieve risk control even when $w$ is unbounded. In particular, assuming $L_i \in [0,B]$, for any potential value $x$ of the covariate, we define

$$\hat{\lambda}(x) = \inf\left\{\lambda : \frac{\sum_{i=1}^n w(X_i)L_i(\lambda) + w(x)B}{\sum_{i=1}^n w(X_i) + w(x)} \leq \alpha\right\}.$$

**Proposition 3.** *In the setting of Theorem 1, with $\hat{\lambda}$ as above,*

$$\mathbb{E}_{(X_1,Y_1),\ldots,(X_n,Y_n)\sim P_{\text{train}},(X_{n+1},Y_{n+1})\sim P_{\text{test}}}[L_{n+1}(\hat{\lambda}(X_{n+1}))] \leq \alpha.$$

This is an exact generalization of the procedure of Tibshirani et al. (2019) beyond indicator losses. As proposed therein, when unlabeled data in the test domain is available, $w$ can be estimated by the probabilistic classification algorithm; this gives good practical results (see also our experiment in Appendix D). For arbitrary distribution shifts, we give a total variation bound analogous to that of Barber et al. (2022) for independent data (see their Section 4.1), though the proof is different. Here we will use the notation $Z_i = (X_i,Y_i)$, and $\hat{\lambda}(Z_1,\ldots,Z_n)$ to refer to that chosen in (4).

**Proposition 4.** *Let $Z = (Z_1,\ldots,Z_{n+1})$ be a sequence of random variables. Then, under the conditions in Theorem 1, $\mathbb{E}\left[L_{n+1}(\hat{\lambda})\right] \leq \alpha + B\sum_{i=1}^n \text{TV}(Z_i,Z_{n+1})$. If further the assumptions of Theorem 2 hold, $\mathbb{E}\left[L_{n+1}(\hat{\lambda})\right] \geq \alpha - B\left(\frac{2}{n+1} + \sum_{i=1}^n \text{TV}(Z_i,Z_{n+1})\right)$.*

### 4.2 QUANTILE RISK CONTROL

Snell et al. (2022) generalizes Bates et al. (2021) to control the quantile of a monotone loss function conditional on $(X_i,Y_i)_{i=1}^n$ with probability $1-\delta$ over the calibration dataset for any user-specified tolerance parameter $\delta$. In some applications, it may be sufficient to control the unconditional quantile of the loss function, which alleviates the burden of the user to choose the tolerance parameter $\delta$.

For any random variable $X$, let $\text{Quantile}_\beta(X) = \inf\{x : \mathbb{P}(X \leq x) \geq \beta\}$. Analogous to (3), we want to find $\hat{\lambda}$ based on $(X_i,Y_i)_{i=1}^n$ such that

$$\text{Quantile}_\beta\left(L_{n+1}(\hat{\lambda}_\beta)\right) \leq \alpha. \tag{11}$$

By definition, $\text{Quantile}_\beta\left(L_{n+1}(\hat{\lambda}_\beta)\right) \leq \alpha \iff \mathbb{E}\left[\mathbb{1}\left\{L_{n+1}(\hat{\lambda}_\beta) > \alpha\right\}\right] \leq 1 - \beta$. As a consequence, quantile risk control is equivalent to expected risk control (3) with loss function $\tilde{L}_i(\lambda) = \mathbb{1}\{L_i(\lambda) > \alpha\}$. Let $\hat{\lambda}_\beta = \inf\left\{\lambda \in \Lambda : \frac{1}{n+1}\sum_{i=1}^n \mathbb{1}\{L_i(\lambda) > \alpha\} + \frac{1}{n+1} \leq 1 - \beta\right\}$.

**Proposition 5.** *In the setting of Theorem 1, with $\hat{\lambda}$ as above, (11) is achieved.*

It is unclear whether the wider class of quantile-based risks considered by Snell et al. (2022) (e.g. the CVaR) can be controlled unconditionally.

### 4.3 CONTROLLING MULTIPLE RISKS

Let $L_i(\lambda; \gamma)$ be a family of loss functions indexed by $\gamma \in \Gamma$ for some domain $\Gamma$ that may have infinitely many elements. A researcher may want to control $\mathbb{E}[L_i(\lambda; \gamma)]$ at level $\alpha(\gamma)$. Equivalently, we need to find an $\hat{\lambda}$ based on $(X_i, Y_i)_{i=1}^n$ such that

$$\sup_{\gamma \in \Gamma} \mathbb{E}\left[\frac{L_i(\hat{\lambda}; \gamma)}{\alpha(\gamma)}\right] \leq 1. \tag{12}$$

Though the above worst-case risk is not an expectation, it can still be controlled. Towards this end, we define $\hat{\lambda} = \sup_{\gamma \in \Gamma} \hat{\lambda}_\gamma$, where $\hat{\lambda}_\gamma = \inf\{\lambda : \frac{1}{n+1}\sum_{i=1}^n L_i(\lambda; \gamma) + \frac{B}{n+1} \leq \alpha(\gamma)\}$.

**Proposition 6.** *In the setting of Theorem 1, with $\hat{\lambda}$ as above, (12) is satisfied.*

### 4.4 ADVERSARIAL RISKS

We next show how to control risks defined by adversarial perturbations. We adopt the same notation as Section 4.3. Bates et al. (2021) (Section 6.3) discusses the adversarial risk where $\Gamma$ parametrizes a class of perturbations of $X_{n+1}$, e.g., $L_i(\lambda; \gamma) = L(X_i + \gamma, Y_i)$ and $\Gamma = \{\gamma : \|\gamma\|_\infty \leq \epsilon\}$. A researcher may want to find an $\hat{\lambda}$ based on $(X_i, Y_i)_{i=1}^n$ such that

$$\mathbb{E}[\sup_{\gamma \in \Gamma} L_i(\lambda; \gamma)] \leq \alpha. \tag{13}$$

This can be recast as a conformal risk control problem by taking $\tilde{L}_i(\lambda) = \sup_{\gamma \in \Gamma} L_i(\lambda; \gamma)$. Then, the following choice of $\lambda$ leads to risk control: $\hat{\lambda} = \inf\{\lambda : \frac{1}{n+1}\sum_{i=1}^n \tilde{L}_i(\lambda) + \frac{B}{n+1} \leq \alpha\}$.

**Proposition 7.** *In the setting of Theorem 1, with $\hat{\lambda}$ as above, (13) is satisfied.*

### 4.5 U-RISK CONTROL

For ranking and metric learning, Bates et al. (2021) considered loss functions that depend on two test points. In general, for any $k > 1$ and subset $\mathcal{S} \subset \{1, \ldots, n+k\}$ with $|\mathcal{S}| = k$, let $L_{\mathcal{S}}(\lambda)$ be a loss function. Our goal is to find $\hat{\lambda}_k$ based on $(X_i, Y_i)_{i=1}^n$ such that

$$\mathbb{E}\left[L_{\{n+1, \ldots, n+k\}}(\hat{\lambda}_k)\right] \leq \alpha. \tag{14}$$

We call the LHS a U-risk since, for any fixed $\hat{\lambda}_k$, it is the expectation of an order-$k$ U-statistic. As a natural extension, we can define

$$\hat{\lambda}_k = \inf\left\{\lambda : \frac{k!n!}{(n+k)!}\sum_{\mathcal{S} \subset \{1, \ldots, n\}, |\mathcal{S}|=k} L_{\mathcal{S}}(\lambda) + B\left(1 - \frac{(n!)^2}{(n+k)!(n-k)!}\right) \leq \alpha\right\}. \tag{15}$$

Again, we define $\hat{\lambda}_k = \lambda_{\max}$ when the RHS is empty. Then we can prove the following result.

**Proposition 8.** *Assume that $L_{\mathcal{S}}(\lambda)$ is non-increasing in $\lambda$, right-continuous, and $L_{\mathcal{S}}(\lambda_{\max}) \leq \alpha$, $\sup_\lambda L_{\mathcal{S}}(\lambda) \leq B < \infty$ almost surely. Then (14) is achieved with $\hat{\lambda}$ as above.*

## 5 CONCLUSION

This generalization of conformal prediction broadens its scope to new applications, as shown in Section 3. Still, two primary limitations of our technique remain: firstly, the requirement of a monotone loss is difficult to lift. Secondly, extensions to non-exchangeable data require knowledge about the form of the shift. This issue affects most statistical methods, including standard conformal prediction, and ours is no different in this regard. Finally, the mathematical tools developed in Sections 2 and 4 may be of independent technical interest, as they provide a new, and more general, language for studying conformal prediction, along with new results about its validity.

### REPRODUCIBILITY STATEMENT

Code to reproduce our examples is available at `https://github.com/aangelopoulos/conformal-risk`.

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

## A    CONFORMAL PREDICTION REDUCES TO RISK CONTROL

Conformal prediction can be thought of as controlling the expectation of an indicator loss function. Recall that the risk upper bound (2) specializes to the conformal coverage guarantee in (1) when the loss function is the indicator of a miscoverage event. The conformal risk control procedure specializes to conformal prediction under this loss function as well. However, the risk lower bound in Theorem 2 has a slightly worse constant than the usual conformal guarantee. We now describe these correspondences.

First, we show the equivalence of the algorithms. In conformal prediction, we have conformal scores $s(X_i, Y_i)$ for some score function $s : \mathcal{X} \times \mathcal{Y} \to \mathbb{R}$. Based on this score function, we create prediction sets for the test point $X_{n+1}$ as $\mathcal{C}_{\hat{\lambda}}(X_{n+1}) = \left\{ y : s(X_{n+1}, y) \leq \hat{\lambda} \right\}$, where $\hat{\lambda}$ is the conformal quantile, a parameter that is set based on the calibration data. In particular, conformal prediction chooses $\hat{\lambda}$ to be the $\lceil (n+1)(1-\alpha) \rceil / n$ sample quantile of $\{s(X_i, Y_i)\}_{i=1}^n$. To formulate this in the language of risk control, we consider a *miscoverage loss* $L_i^{\mathrm{Cvg}}(\lambda) = \mathbb{1}\left\{ Y_i \notin \widehat{\mathcal{C}}_\lambda(X_i) \right\} = \mathbb{1}\left\{ s(X_i, Y_i) > \lambda \right\}$. Direct calculation of $\hat{\lambda}$ from (4) then shows the equivalence of the proposed procedure to conformal prediction:

$$
\hat{\lambda} = \inf \left\{ \lambda : \frac{1}{n+1} \sum_{i=1}^n \mathbb{1}\left\{ s(X_i, Y_i) > \lambda \right\} + \frac{1}{n+1} \leq \alpha \right\} =
$$

$$
\underbrace{\inf \left\{ \lambda : \frac{1}{n} \sum_{i=1}^n \mathbb{1}\left\{ s(X_i, Y_i) \leq \lambda \right\} \geq \frac{\lceil (n+1)(1-\alpha) \rceil}{n} \right\}}_{\text{conformal prediction algorithm}}.
$$

Next, we discuss how the risk lower bound relates to its conformal prediction equivalent. In the setting of conformal prediction, Lei et al. (2018) proves that $\mathbb{P}(Y_{n+1} \notin \mathcal{C}_{\hat{\lambda}}(X_{n+1})) \geq \alpha - 1/(n+1)$ when the conformal score function follows a continuous distribution. Theorem 2 recovers this guarantee with a slightly worse constant: $\mathbb{P}(Y_{n+1} \notin \mathcal{C}_{\hat{\lambda}}(X_{n+1})) \geq \alpha - 2/(n+1)$. First, note that our

assumption in Theorem 2 about the distribution of discontinuities specializes to the continuity of the score function when the miscoverage loss is used: $\mathbb{P}\left(J\left(L_i^{\mathrm{Cvg}}, \lambda\right) > 0\right) = 0 \iff \mathbb{P}(s(X_i, Y_i) = \lambda) = 0$. However, the bound for the conformal case is better than the bound for the general case in Theorem 2 by a factor of two, which cannot be improved according to Proposition 1. This difference is an interesting oddity of the binary loss, but is not practically important.

## B  COMPARISON WITH RCPS/LTT

The setting of conformal risk control resembles the setting of previous works in high-probability risk control Bates et al. (2021); Angelopoulos et al. (2021a) (RCPS/LTT); however, the algorithms, guarantees, and proof strategies are entirely different.

To summarize, the main differences are as follows:

1. Conformal risk control is substantially more sample-efficient than RCPS/LTT. On the scale of the risk, RCPS/LTT are far more conservative, converging to $\alpha$ at a $\frac{1}{\sqrt{n}}$ rate, while conformal risk control converges at a $\frac{1}{n}$ rate.

2. RCPS/LTT are high probability bounds of the form

$$\mathbb{P}(\mathbb{E}[\ell(Y_{n+1}, \mathcal{C}_{\hat{\lambda}}(X_{n+1})) \mid \{X_i, Y_i\}_{i=1}^n] \leq \alpha) \geq 1 - \delta$$

Conformal risk control gives an expectation bound of the form

$$\mathbb{E}[\ell(Y_{n+1}, \mathcal{C}_{\hat{\lambda}}(X_{n+1}))] \leq \alpha.$$

The former bound provides high-probability control over the sampling of the calibration data, which can be attractive if one needs to be *below* $\alpha$ with high probability, as opposed to *centered at* $\alpha$ as in the latter. However, the former bound is more difficult to interpret for nonstatisticians and requires the selection of two parameters—$\alpha$ and $\delta$—which can introduce complexity.

3. RCPS/LTT are substantially more complicated procedures than conformal risk control— *especially* LTT, which requires careful discretization and bespoke multiple testing correction. This careful construction is required because the LTT procedure allows for the high-probability control of non-monotone risks, while conformal risk control only applies to monotone ones.

4. The theory of conformal risk control is a basic advancement to the core theory of conformal prediction and exchangeability-based arguments, extending them to a broader mathematical setting. Meanwhile, LTT/RCPS use well-established concentration bounds, which are mathematically unrelated to conformal prediction.

To underscore these points, we perform a careful evaluation against these procedures on the polyp segmentation dataset in Figure 5. We compare against two versions of LTT/RCPS: Baseline 1 has $\delta = 0.5$, Baseline 2 integrates the tail bound of LTT/RCPS to achieve a bound in expectation. The former strategy is not statistically valid for the goal of expectation control, and the latter strategy is essentially only possible using fixed-width bounds such as Hoeffding's inequality, and is described in the below Appendix B.1. We use the same risk level $\alpha = 0.1$ for all procedures and make plots with $n = \{25, 50, 100, 200, 300\}$.

The takeaways are as follows: The statistical efficiency gains of conformal risk control are massive. On the scale of the risk gap, $|\mathbb{E}[R(\hat{\lambda})] - \alpha|$, conformal risk control is 50% closer to the desired risk than Baseline 1, and orders of magnitude closer than Baseline 2. This is expected given the theoretical convergence rate is quadratically faster than both approaches. The computational efficiency of conformal risk control is substantially better than all baselines. For example, it is 14x faster than Baseline 1 and 19% faster than Baseline 2. It is also substantially easier to implement than Baseline 1, only requiring 5 lines of code. Baseline 2 is equally easy to implement, although far less flexible.

### B.1  RECOVERING EXPECTATION BOUNDS FROM RCPS

We introduce a technique for deriving bounds in expectation using RCPS Bates et al. (2021)/LTT Angelopoulos et al. (2021a). This technique was not described in the original papers;

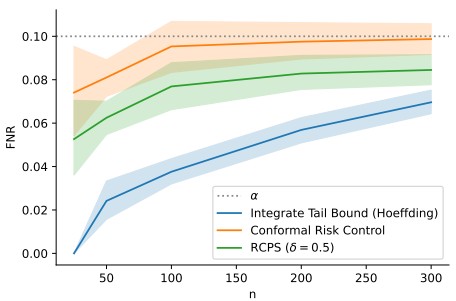 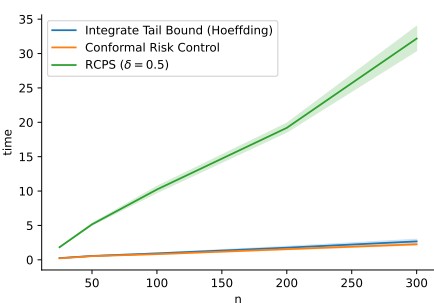

Figure 5: **Comparison of RCPS/LTT with conformal risk control on the polyp dataset.**

we only derive it here for the purpose of the above comparison (i.e., to show that this approach should essentially never be used for monotone losses).

Consider a pointwise upper-bound on the empirical risk, as constructed in RCPS Bates et al. (2021):

$$\mathbb{P}(R(\lambda) - \hat{R}_n(\lambda) > \epsilon) < \delta(\epsilon, \lambda),$$

for all $\epsilon \in [0, B]$. The next proposition says that if the bound has equal width everywhere, like in the case of Hoeffding's inequality, then it can be integrated to control the risk. Unfortunately, this strategy is *not* possible for general concentration inequalities, such as that of Bentkus (2004) or Waudby-Smith & Ramdas (2020), as it would require knowing $\delta(\epsilon, \lambda^*)$ (for an oracle $\lambda^*$ defined in the proof below).

**Proposition B.1.** *Let $L_i(\lambda)$ be right-continuous, monotone, i.i.d. functions in $[0, B]$ satisfying $\mathbb{P}(J(L_i, \lambda) > 0) = 0$. Furthermore, define $\delta^+(\epsilon) = \sup_\lambda \delta(\epsilon, \lambda)$ and*

$$\hat{\lambda} = \inf \left\{ \lambda : \hat{R}(\lambda) \leq \alpha^- \right\},$$

*where $\alpha^-$ satisfies $\alpha^- \leq \alpha - \int_0^{B-\alpha^-} \delta^+(\epsilon) d\epsilon$. Then,*

$$R(\hat{\lambda}) \leq \alpha.$$

*Proof.* Let $\lambda^* = \inf\{\lambda : R(\lambda) \leq \alpha^- + \epsilon\}$. By monotonicity and the definition of $\hat{\lambda}$,

$$\hat{\lambda} \geq \lambda^* \implies \hat{R}_n(\lambda^*) \leq \alpha^-.$$

But also, by the bounded jump assumption, $R(\lambda^*) = \alpha^- + \epsilon$. Therefore, on the event $\hat{\lambda} \geq \lambda^*$, we have that

$$R(\lambda^*) - \hat{R}_n(\lambda^*) \geq \alpha^- - \alpha^- + \epsilon = \epsilon. \tag{16}$$

Since the event in (16) happens with probability at most $\delta^+(\epsilon)$, we have that $\mathbb{P}(\hat{\lambda} \geq \lambda^*) \leq \delta^+(\epsilon)$, so

$$\mathbb{P}(R(\hat{\lambda}) \geq \alpha^- + \epsilon) \leq \delta^+(\epsilon).$$

Finally, we have that

$$\mathbb{E}[R(\hat{\lambda})] = \int_0^B \mathbb{P}(R(\hat{\lambda}) > r) dr$$

$$\leq \alpha^- + \int_{\alpha^-}^B \mathbb{P}(R(\hat{\lambda}) > \alpha^- + \epsilon) d\epsilon$$

$$\leq \alpha^- + \int_0^{B-\alpha^-} \delta^+(\epsilon) d\epsilon$$

$$\leq \alpha.$$

$\square$

This can be applied to Hoeffding's inequality, giving the aforementioned Baseline 2.

**Corollary 2.** *In the setting of Proposition B.1, let* $\alpha^- = \alpha - \frac{\sqrt{\pi} B \; \mathrm{erf}(\sqrt{2n}/B)}{2\sqrt{2n}}$. *Then,* $R(\hat{\lambda}) \leq \alpha$.

*Proof.* Applying Hoeffding's inequality we have $\delta^+(\epsilon) = e^{-2n\epsilon^2}$, and then

$$\alpha - \int_0^{1-\alpha^-} e^{-\frac{2n\epsilon^2}{B^2}} d\epsilon \geq \alpha - \int_0^1 e^{-\frac{2n\epsilon^2}{B^2}} d\epsilon = \alpha - \frac{\sqrt{\pi} B \; \mathrm{erf}(\sqrt{2n}/B)}{2\sqrt{2n}} = \alpha^-,$$

as required. □

## C  MONOTONIZING NON-MONOTONE RISKS

We next show that the proposed algorithm leads to asymptotic risk control for non-monotone risk functions when applied to a monotonized version of the empirical risk. We set the *monotonized empirical risk* to be

$$\widehat{R}_n^\uparrow(\lambda) = \sup_{t \geq \lambda} \widehat{R}_n(t),$$

then define

$$\hat{\lambda}_n^\uparrow = \inf \left\{ \lambda : \widehat{R}_n^\uparrow(\lambda) \leq \alpha \right\}.$$

**Theorem C.1.** *Let the $L_i(\lambda)$ be right-continuous, i.i.d., bounded (both above and below) functions satisfying* (5). *Then,*

$$\lim_{n \to \infty} \mathbb{E}\left[ L_{n+1}(\hat{\lambda}_n^\uparrow) \right] \leq \alpha.$$

Theorem C.1 implies that an analogous procedure to 4 also controls the risk asymptotically. In particular, taking

$$\tilde{\lambda}^\uparrow = \inf \left\{ \lambda : \widehat{R}_n^\uparrow(\lambda) + \frac{B}{n+1} \leq \alpha \right\}$$

also results in asymptotic risk control (to see this, plug $\tilde{\lambda}^\uparrow$ into Theorem C.1 and see that the risk level is bounded above by $\alpha - \frac{B}{n+1}$). Note that in the case of a monotone loss function, $\tilde{\lambda}^\uparrow = \hat{\lambda}$. However, the counterexample in Proposition 2 does not apply to $\tilde{\lambda}^\uparrow$, and it is currently unknown whether this procedure does or does not provide finite-sample risk control.

## D  EXPERIMENTS ON COVARIATE SHIFT

To validate the covariate shift extension from Section 4, we perform an experiment on a synthetic regression under covariate shift. The model is as follows:

$$X_i \sim \mathcal{N}(0, 1)$$
$$Y_i = 2X_i + \mathcal{N}(0, 0.5|X_i|)$$
$$X_{\mathrm{test}} \sim \mathcal{N}(0, 2)$$
$$Y_{\mathrm{test}} = 2X_{\mathrm{test}} + \mathcal{N}(0, 0.5|X_{\mathrm{test}}|).$$

The model used for prediction was a standard linear regression pre-trained on 1000 data points independent and identically distributed with the calibration data. We estimated the likelihood ratio using a logistic regression model by solving a classification problem to classify between all the available training/calibration covariates and batch of 1000 unlabeled test datapoints from the shifted distribution. The probits from the logistic regression were then transformed to likelihood ratios via the function $h(x) = x/1 - x$. We took $n = 100$ and $\alpha = 0.1$, controlling the clipped projective distance of $y$ onto the set $C$:

$$\ell(y, C) = \min(\mathrm{proj}_C(y), B),$$

where $B = 2$. The set construction is the standard

$$\mathcal{C}_\lambda(x) = \left[ \hat{Y}(x) \pm \lambda \right].$$

Figure 6 shows results.

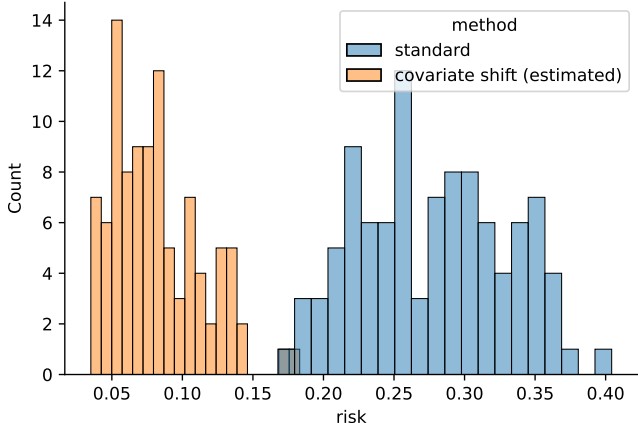

Figure 6: **Risk control results on a synthetic covariate shift dataset.** Running standard conformal risk control does not control the risk, while running the covariate shift algorithm from Section 4 does—even though the likelihood ratios are estimated by a logistic regression model. We also computed the risk with the true likelihood ratios, and it overlaps entirely with the histogram for estimated likelihood ratios, so we do not show it here.

## E   PROOFS

The proof of Theorem 2 uses the following lemma on the approximate continuity of the empirical risk.

**Lemma 1** (Jump Lemma). *In the setting of Theorem 2, any jumps in the empirical risk are bounded, i.e.,*

$$\sup_\lambda J\big(\widehat{R}_n, \lambda\big) \overset{\text{a.s.}}{\leq} \frac{B}{n}.$$

*Proof of Jump Lemma, Lemma 1.* By boundedness, the maximum contribution of any single point to the jump is $\frac{B}{n}$, so

$$\exists \lambda : \ J\big(\widehat{R}_n, \lambda\big) > \frac{B}{n} \implies \exists \lambda : \ J(L_i, \lambda) > 0 \text{ and } J(L_j, \lambda) > 0 \text{ for some } i \neq j.$$

Call $\mathcal{D}_i = \{\lambda : J(L_i, \lambda) > 0\}$ the sets of discontinuities in $L_i$. Since $L_i$ is bounded monotone, $\mathcal{D}_i$ has countably many points. The union bound then implies that

$$\mathbb{P}\left(\exists \lambda : \ J(\widehat{R}_n, \lambda) > \frac{B}{n}\right) \leq \sum_{i \neq j} \mathbb{P}(\mathcal{D}_i \cap \mathcal{D}_j \neq \emptyset)$$

Rewriting each term of the right-hand side using tower property and law of total probability gives

$$\mathbb{P}\left(\mathcal{D}_i \cap \mathcal{D}_j \neq \emptyset\right) = \mathbb{E}\left[\mathbb{P}\big(\mathcal{D}_i \cap \mathcal{D}_j \neq \emptyset \,\big|\, \mathcal{D}_j\big)\right]$$

$$\leq \mathbb{E}\left[\sum_{\lambda \in \mathcal{D}_j} \mathbb{P}\left(\lambda \in \mathcal{D}_i \,\Big|\, \mathcal{D}_j\right)\right] = \mathbb{E}\left[\sum_{\lambda \in \mathcal{D}_j} \mathbb{P}\left(\lambda \in \mathcal{D}_i\right)\right],$$

Where the second inequality is because the union of the events $\lambda \in \mathcal{D}_j$ is the entire sample space, but they are not disjoint, and the third equality is due to the independence between $\mathcal{D}_i$ and $\mathcal{D}_j$. Rewriting in terms of the jump function and applying the assumption $\mathbb{P}\left(J(L_i, \lambda) > 0\right) = 0$,

$$\mathbb{E}\left[\sum_{\lambda \in \mathcal{D}_j} \mathbb{P}\left(\lambda \in \mathcal{D}_i\right)\right] = \mathbb{E}\left[\sum_{\lambda \in \mathcal{D}_j} \mathbb{P}\left(J(L_i, \lambda) > 0\right)\right] = 0.$$

Chaining the above inequalities yields $\mathbb{P}\left(\exists\lambda : J(\widehat{R}_n, \lambda) > \frac{B}{n}\right) \leq 0$, so

$$\mathbb{P}\left(\exists\lambda : J(\widehat{R}_n, \lambda) > \frac{B}{n}\right) = 0. \qquad \square$$

*Proof of Theorem 2.* If $L_i(\lambda_{\max}) \geq \alpha - 2B/(n+1)$, then $\mathbb{E}[L_{n+1}(\hat{\lambda})] \geq \alpha - 2B/(n+1)$. Throughout the rest of the proof, we assume that $L_i(\lambda_{\max}) < \alpha - 2B/(n+1)$. Define the quantity

$$\hat{\lambda}'' = \inf\left\{\lambda : \widehat{R}_{n+1}(\lambda) + \frac{B}{n+1} \leq \alpha\right\}.$$

Since $L_i(\lambda_{\max}) < \alpha - 2B/(n+1) < \alpha - B/(n+1)$, $\hat{\lambda}''$ exists almost surely. Deterministically, $\frac{n}{n+1}\widehat{R}_n(\lambda) \leq \widehat{R}_{n+1}(\lambda)$, which yields $\hat{\lambda} \leq \hat{\lambda}''$. Again since $L_i(\lambda)$ is non-increasing in $\lambda$,

$$\mathbb{E}\left[L_{n+1}(\hat{\lambda}'')\right] \leq \mathbb{E}\left[L_{n+1}(\hat{\lambda})\right]$$

By exchangeability and the fact that $\hat{\lambda}''$ is a symmetric function of $L_1, \ldots, L_{n+1}$,

$$\mathbb{E}\left[L_{n+1}(\hat{\lambda}'')\right] = \mathbb{E}\left[\widehat{R}_{n+1}(\hat{\lambda}'')\right]$$

For the remainder of the proof we focus on lower-bounding $\widehat{R}_{n+1}(\hat{\lambda}'')$. We begin with the following identity:

$$\alpha = \widehat{R}_{n+1}(\hat{\lambda}'') + \frac{B}{n+1} - \left(\widehat{R}_{n+1}(\hat{\lambda}'') + \frac{B}{n+1} - \alpha\right).$$

Rearranging the identity,

$$\widehat{R}_{n+1}(\hat{\lambda}'') = \alpha - \frac{B}{n+1} + \left(\widehat{R}_{n+1}(\hat{\lambda}'') + \frac{B}{n+1} - \alpha\right).$$

Using the Jump Lemma to bound $\left(\widehat{R}_{n+1}(\hat{\lambda}'') + \frac{B}{n+1} - \alpha\right)$ below by $-\frac{B}{n+1}$ gives

$$\widehat{R}_{n+1}(\hat{\lambda}'') \geq \alpha - \frac{2B}{n+1}.$$

Finally, chaining together the above inequalities,

$$\mathbb{E}\left[L_{n+1}(\hat{\lambda})\right] \geq \mathbb{E}\left[\widehat{R}_{n+1}(\hat{\lambda}'')\right] \geq \alpha - \frac{2B}{n+1}.$$

$$\square$$

*Proof of Proposition 1.* Without loss of generality, assume $B = 1$. Fix any $\epsilon' > 0$. Consider the following loss functions, which satisfy the conditions in Theorem 2:

$$L_i(\lambda) \overset{i.i.d.}{\sim} \begin{cases} 1 & \lambda \in [0, Z_i) \\ \frac{k}{k+1} & \lambda \in [Z_i, W_i) \\ 0 & \text{else} \end{cases},$$

where $k \in \mathbb{N}$, the $Z_i \overset{i.i.d.}{\sim} \text{Uniform}(0, 0.5)$, the $W_i \overset{i.i.d.}{\sim} \text{Uniform}(0.5, 1)$ for $i \in \{1, ..., n+1\}$ and $\alpha = \frac{k+1-\epsilon'}{n+1}$. Then, by the definition of $\hat{\lambda}$, we know

$$\widehat{R}_n(\hat{\lambda}) \leq \frac{k - \epsilon'}{n}. \tag{17}$$

If $n > k + 1$, $\widehat{R}(\lambda) \geq \frac{k}{k+1} > \frac{k}{n}$ whenever $\lambda \leq \frac{1}{2}$. Thus, we must have $\hat{\lambda} > \frac{1}{2}$. Since $k$ is an integer and by (17), we know that $\left|\{i \in \{1, ..., n\} : L_i(\hat{\lambda}) > 0\}\right| \leq \lfloor(k+1)(k-\epsilon')/k\rfloor \leq k$. This immediately implies that

$$\hat{\lambda} \geq W_{(n-k+1)},$$

where $W_{(j)}$ denotes the $j$-th order statistic. Notice that for all $\lambda > \frac{1}{2}$,

$$R(\lambda) = \mathbb{E}\left[L_i(\lambda)\right] = \frac{k}{k+1}\mathbb{P}(W_i > \lambda) = \frac{k}{k+1} \cdot 2(1 - \lambda),$$

so $R(\hat{\lambda}) \leq \frac{k}{k+1} \cdot 2(1 - W_{(n-k+1)})$. Let $U_{(k)}$ be the $k$-th smallest order statistic of $n$ i.i.d. uniform random variables on $(0, 1)$. Then, by symmetry and rescaling, $2(1 - W_{(n-k+1)}) \overset{d}{=} U_{(k)}$,

$$R(\hat{\lambda}) \preceq \frac{k}{k+1}U_{(k)},$$

where $\preceq$ denotes the stochastic dominance. It is well-known that $U_{(k)} \sim \text{Beta}(k, n+1-k)$ and hence

$$\mathbb{E}[R(\hat{\lambda})] \leq \frac{k}{k+1} \cdot \frac{k}{n+1}.$$

Thus,

$$\alpha - \mathbb{E}\left[R(\hat{\lambda})\right] \geq \frac{k+1-\epsilon}{n+1} - \frac{k^2}{(n+1)(k+1)} = \frac{1}{n+1} \cdot \frac{(2-\epsilon')k + 1 - \epsilon'}{k+1}.$$

For any given $\epsilon > 0$, let $\epsilon' = \epsilon/2$ and $k = \lceil \frac{2}{\epsilon} - 1 \rceil$. Then

$$\frac{(2 - \epsilon')k + 1 - \epsilon'}{k+1} \geq 2 - \epsilon,$$

implying that

$$\alpha - \mathbb{E}\left[R(\hat{\lambda})\right] \geq \frac{2 - \epsilon}{n+1}.$$

$\square$

*Proof of Proposition 2.* Without loss of generality, we assume $B = 1$. Assume $\hat{\lambda}$ takes values in $[0, 1]$ and $\alpha \in (1/(n+1), 1)$. Let $p \in (0, 1)$, $N$ be any positive integer, and $L_i(\lambda)$ be i.i.d. right-continuous piecewise constant (random) functions with

$$L_i(N/N) = 0, \quad (L_i(0/N), L_i(1/N), \ldots, L_i((N-1)/N)) \overset{i.i.d.}{\sim} \text{Ber}(p).$$

By definition, $\hat{\lambda}$ is independent of $L_{n+1}$. Thus, for any $j = 0, 1, \ldots, N-1$,

$$\left\{L_{n+1}(\hat{\lambda}) \mid \hat{\lambda} = j/N\right\} \sim \text{Ber}(p), \quad \left\{L_{n+1}(\hat{\lambda}) \mid \hat{\lambda} = 1\right\} \sim \delta_0.$$

Then,

$$\mathbb{E}\left[L_{n+1}(\hat{\lambda})\right] = p \cdot \mathbb{P}(\hat{\lambda} \neq 1)$$

Note that

$$\hat{\lambda} \neq 1 \iff \min_{j \in \{0, \ldots, N-1\}} \frac{1}{n+1}\sum_{i=1}^{n} L_i(j/N) \leq \alpha - \frac{1}{n+1}.$$

Since $\alpha > 1/(n+1)$,

$$\mathbb{P}(\hat{\lambda} \neq 1) = 1 - \mathbb{P}(\hat{\lambda} = 1) = 1 - \mathbb{P}\left(\text{for all } j, \text{ we have } \frac{1}{n+1}\sum_{i=1}^{n} L_i(j/N) > \alpha - \frac{1}{n+1}\right)$$

$$= 1 - \left(\sum_{k=\lceil(n+1)\alpha\rceil}^{n} \binom{n}{k}p^k(1-p)^{(n-k)}\right)^N$$

$$= 1 - \left(1 - \text{BinoCDF}\left(n, p, \lceil(n+1)\alpha\rceil - 1\right)\right)^N$$

As a result,

$$\mathbb{E}\left[L_{n+1}(\hat{\lambda})\right] = p\left(1 - \left(1 - \text{BinoCDF}\left(n, p, \lceil(n+1)\alpha\rceil - 1\right)\right)^N\right).$$

Now let $N$ be sufficiently large such that

$$\left(1 - \left(1 - \text{BinoCDF}\big(n, p, \lceil (n+1)\alpha \rceil - 1\big)\right)^N\right) > p.$$

Then

$$\mathbb{E}\left[L_{n+1}(\hat{\lambda})\right] > p^2$$

For any $\alpha > 0$, we can take $p$ close enough to 1 to render the claim false. $\qquad\square$

*Proof of Theorem C.1.* Define the *monotonized population risk* as

$$R^\uparrow(\lambda) = \sup_{t \geq \lambda} \mathbb{E}\left[L_{n+1}(t)\right]$$

Note that the independence of $L_{n+1}$ and $\hat{\lambda}_n^\uparrow$ implies that for all $n$,

$$\mathbb{E}\left[L_{n+1}\big(\hat{\lambda}_n^\uparrow\big)\right] \leq \mathbb{E}\left[R^\uparrow\big(\hat{\lambda}_n^\uparrow\big)\right].$$

Since $R^\uparrow$ is bounded, monotone, and one-dimensional, a generalization of the Glivenko-Cantelli Theorem given in Lemma 2 gives that uniformly over $\lambda$,

$$\lim_{n \to \infty} \sup_\lambda |\widehat{R}_n(\lambda) - R(\lambda)| \overset{\text{a.s.}}{\to} 0.$$

As a result,

$$\lim_{n \to \infty} \sup_\lambda |\widehat{R}_n^\uparrow(\lambda) - R^\uparrow(\lambda)| \overset{\text{a.s.}}{\to} 0,$$

which implies that

$$\lim_{n \to \infty} |\widehat{R}_n^\uparrow(\hat{\lambda}^\uparrow) - R^\uparrow(\hat{\lambda}^\uparrow)| \overset{\text{a.s.}}{\to} 0.$$

By definition, $\widehat{R}^\uparrow(\hat{\lambda}^\uparrow) \leq \alpha$ almost surely and thus this directly implies

$$\limsup_{n \to \infty} R^\uparrow\big(\hat{\lambda}_n^\uparrow\big) \leq \alpha \quad \text{a.s..}$$

Finally, since for all $n$, $R^\uparrow\big(\hat{\lambda}_n^\uparrow\big) \leq B$, by Fatou's lemma,

$$\lim_{n \to \infty} \mathbb{E}\left[L_{n+1}\big(\hat{\lambda}_n^\uparrow\big)\right] \leq \limsup_{n \to \infty} \mathbb{E}\left[R^\uparrow\big(\hat{\lambda}_n^\uparrow\big)\right] \leq \mathbb{E}\left[\limsup_{n \to \infty} R^\uparrow\big(\hat{\lambda}_n^\uparrow\big)\right] \leq \alpha.$$

$$\square$$

*Proposition 3.* Let

$$\hat{\lambda}' = \inf\left\{\lambda : \frac{\sum_{i=1}^{n+1} w(X_i) L_i(\lambda)}{\sum_{i=1}^{n+1} w(X_i)} \leq \alpha\right\}.$$

Since $\inf_\lambda L_i(\lambda) \leq \alpha$, $\hat{\lambda}'$ exists almost surely. Using the same argument as in the proof of Theorem 1, we can show that $\hat{\lambda}' \leq \hat{\lambda}(X_{n+1})$. Since $L_{n+1}(\lambda)$ is non-increasing in $\lambda$,

$$\mathbb{E}[L_{n+1}(\hat{\lambda}(X_{n+1}))] \leq \mathbb{E}[L_{n+1}(\hat{\lambda}')].$$

Let $E$ be the multiset of loss functions $\{(X_1, Y_1), \ldots, (X_{n+1}, Y_{n+1})\}$. Then $\hat{\lambda}'$ is a function of $E$, or, equivalently, $\hat{\lambda}'$ is a constant conditional on $E$. Lemma 3 of Tibshirani et al. (2019) implies that

$$(X_{n+1}, Y_{n+1}) \mid E \sim \sum_{i=1}^{n+1} \frac{w(X_i)}{\sum_{j=1}^{n+1} w(X_j)} \delta_{(X_j, Y_j)} \implies L_{n+1} \mid E \sim \sum_{i=1}^{n+1} \frac{w(X_i)}{\sum_{j=1}^{n+1} w(X_j)} \delta_{L_i}$$

where $\delta_z$ denotes the Dirac measure at $z$. Together with the right-continuity of $L_i$, the above result implies

$$\mathbb{E}\left[L_{n+1}(\hat{\lambda}') \mid E\right] = \frac{\sum_{i=1}^{n+1} w(X_i) L_i(\hat{\lambda}')}{\sum_{i=1}^{n+1} w(X_i)} \leq \alpha.$$

The proof is then completed by the law of total expectation. $\qquad\square$

**Lemma 2** (Glivenko-Cantelli for Monotone Functions). *For i.i.d., right-continuous, monotone loss functions $L_1, \ldots, L_n$, we have that*

$$\lim_{n \to \infty} \sup_{\lambda} |\widehat{R}_n(\lambda) - R(\lambda)| \overset{a.s.}{=} 0.$$

*Lemma 2.* We generalize the proof of the Glivenko-Cantelli Theorem (see, e.g., Van der Vaart (2000)). Let the notation $R(\lambda-)$ denote the limit of $R$ from the left-hand side approaching $\lambda$. By the strong law of large numbers, $\widehat{R}_n(\lambda) \overset{a.s.}{\to} R(\lambda)$ and $\widehat{R}_n(\lambda-) \overset{a.s.}{\to} R(\lambda-)$ for all $\lambda$. Fix $\epsilon > 0$. It is well-known that a monotone function cannot have more than a countable number of discontinuities (Froda, 1931). Thus, there exists a finite partition $\Lambda(\epsilon) = \{-\infty \le \lambda_1 \le \ldots \le \lambda_k \le \infty\}$, such that $R(\lambda_i-) - R(\lambda_{i-1}) < \epsilon$. (There can only be a finite number of jumps greater than $\epsilon$ in size, and these can be included in $\Lambda(\epsilon)$.) The uniform convergence on the finite set $\Lambda(\epsilon)$ is immediate by the law of large numbers; thus, we have that $\lim_{n \to \infty} \sup_{\lambda \in \Lambda(\epsilon)} |\widehat{R}_n(\lambda) - R(\lambda)| \overset{a.s.}{=} 0$. For any $\lambda \notin \Lambda(\epsilon)$, suppose $\lambda \in (\lambda_{j-1}, \lambda_j)$. Then $\hat{R}_n(\lambda) - R(\lambda) \le \hat{R}_n(\lambda_j) - R(\lambda_{j-1}) < \hat{R}_n(\lambda_j) - R(\lambda_j) + \epsilon$. Similarly, $\hat{R}_n(\lambda) - R(\lambda) \ge \hat{R}_n(\lambda_{j-1}) - R(\lambda_j^-) > \hat{R}_n(\lambda_{j-1}) - R(\lambda_{j-1}) - \epsilon$. Combining two pieces, we conclude that $\sup_{\lambda} |\hat{R}_n(\lambda) - R(\lambda)| \le \sup_{\lambda \in \Lambda(\epsilon)} |\widehat{R}_n(\lambda) - R(\lambda)| + \epsilon$. As a result, $\lim_{n \to \infty} \sup_{\lambda} |\widehat{R}_n(\lambda) - R(\lambda)| \overset{a.s.}{\le} \epsilon$. Taking $\epsilon \to 0$ completes the proof. $\qquad\square$

*Proposition 4.* Define the vector $Z' = (Z'_1, \ldots, Z'_n, Z_{n+1})$, where $Z'_i \overset{i.i.d.}{\sim} \mathcal{L}(Z_{n+1})$ for all $i \in [n]$. Let

$$\epsilon = \sum_{i=1}^{n} \mathrm{TV}(Z_i, Z'_i).$$

By sublinearity,

$$\mathrm{TV}(Z, Z') \le \epsilon. \tag{18}$$

It is a standard fact that (18) implies

$$\sup_{f \in \mathcal{F}_1} |\mathbb{E}[f(Z)] - \mathbb{E}[f(Z')]| \le \epsilon,$$

where $\mathcal{F}_1 = \{f : \mathcal{Z} \mapsto [0, 1]\}$. Let $\ell : \mathcal{Z} \times \Lambda \to [0, B]$ be a bounded loss function. Furthermore, let $g(z) = \ell(z_{n+1}; \hat{\lambda}(z_1, \ldots, z_n))$. Since $g(Z) \in [0, B]$,

$$|\mathbb{E}[g(Z)] - \mathbb{E}[g(Z')]| \le B\epsilon.$$

Furthermore, since $Z'_1, \ldots, Z'_{n+1}$ are exchangeable, we can apply Theorems 1 and 2 to $\mathbb{E}[g(Z')]$, recovering

$$\alpha - \frac{2B}{n+1} \le \mathbb{E}[g(Z')] \le \alpha.$$

A final step of triangle inequality implies the result:

$$\alpha - \frac{2B}{n+1} - B\epsilon \le \mathbb{E}[g(Z)] \le \alpha + B\epsilon.$$

$$\square$$

*Proposition 5.* It is left to prove that $\tilde{L}_i(\lambda)$ satisfies the conditions of Theorem 1. It is clear that $\tilde{L}_i(\lambda) \le 1$ and $\tilde{L}_i(\lambda)$ is non-increasing in $\lambda$ when $L_i(\lambda)$ is. Since $L_i(\lambda)$ is non-increasing and right-continuous, for any sequence $\lambda_m \downarrow \lambda$,

$$L_i(\lambda_m) \uparrow L_i(\lambda) \implies \mathbb{1}\{L_i(\lambda_m) > \alpha\} \to \mathbb{1}\{L_i(\lambda) > \alpha\}.$$

Thus, $\tilde{L}_i(\lambda)$ is right-continuous. Finally, $L_i(\lambda_{\max}) \le \alpha$ implies $\tilde{L}_i(\lambda_{\max}) = 0 \le 1 - \beta$. $\qquad\square$

*Proposition 6.* Examining the form of $\hat{\lambda}$, for each $\gamma \in \Gamma$, we have

$$\mathbb{E}\left[L(\hat{\lambda}, \gamma)\right] \le \mathbb{E}\left[L(\hat{\lambda}_\gamma, \gamma)\right] \le \alpha(\gamma).$$

Thus, dividing both sides by $\alpha(\gamma)$ and taking the supremum, we get that $\sup_{\gamma \in \Gamma} \mathbb{E}\left[\frac{L(\hat{\lambda}, \gamma)}{\alpha(\gamma)}\right] \le 1$, and the worst-case risk is controlled. $\qquad\square$

*Proposition 7.* Because $L_i(\lambda, \gamma)$ is bounded and monotone in $\lambda$ for all choices of $\gamma$, it is also true that $\tilde{L}_i(\lambda)$ is bounded and monotone. Furthermore, the pointwise supremum of right-continuous functions is also right-continuous. Therefore, the $\tilde{L}_i$ satisfy the assumptions of Theorem 1. $\square$

*Proposition 8.* Let

$$\hat{\lambda}'_k = \inf \left\{ \lambda : \frac{k!n!}{(n+k)!} \sum_{\mathcal{S} \subset \{1,\ldots,n+k\}, |\mathcal{S}|=k} L_{\mathcal{S}}(\lambda) \leq \alpha \right\}.$$

Since $L_{\mathcal{S}}(\lambda_{\max}) \leq \alpha$, $\hat{\lambda}'_k$ exists almost surely. Since $L_{\mathcal{S}}(\lambda) \leq B$, we have

$$\frac{k!n!}{(n+k)!} \sum_{\mathcal{S} \subset \{1,\ldots,n+k\}, |\mathcal{S}|=k} L_{\mathcal{S}}(\lambda)$$

$$\leq \frac{k!n!}{(n+k)!} \sum_{\mathcal{S} \subset \{1,\ldots,n\}, |\mathcal{S}|=k} L_{\mathcal{S}}(\lambda) + B \cdot \sum_{\mathcal{S} \cap \{n+1,\ldots,n+k\} \neq \emptyset, |\mathcal{S}|=k} 1$$

$$= \frac{k!n!}{(n+k)!} \sum_{\mathcal{S} \subset \{1,\ldots,n\}, |\mathcal{S}|=k} L_{\mathcal{S}}(\lambda) + B \left( 1 - \frac{k!n!}{(n+k)!} \sum_{\mathcal{S} \subset \{1,\ldots,n\}, |\mathcal{S}|=k} 1 \right)$$

$$= \frac{k!n!}{(n+k)!} \sum_{\mathcal{S} \subset \{1,\ldots,n\}, |\mathcal{S}|=k} L_{\mathcal{S}}(\lambda) + B \left( 1 - \frac{(n!)^2}{(n+k)!(n-k)!} \right).$$

Since $L_{\mathcal{S}}(\lambda)$ is non-increasing in $\lambda$, we conclude that $\hat{\lambda}'_k \leq \hat{\lambda}_k$ if the right-hand side of (15) is not empty; otherwise, by definition, $\hat{\lambda}'_k \leq \lambda_{\max} = \hat{\lambda}_k$. Thus, $\hat{\lambda}'_k \leq \hat{\lambda}_k$ almost surely. Let $E$ be the multiset of loss functions $\{L_{\mathcal{S}} : \mathcal{S} \subset \{1,\ldots,n+k\}, |\mathcal{S}|=k\}$. Using the same argument in the end of the proof of Theorem 1 and the right-continuity of $L_{\mathcal{S}}$, we can show that

$$\mathbb{E}\left[ L_{\{n+1,\ldots,n+k\}}(\hat{\lambda}'_k) \mid E \right] = \frac{k!n!}{(n+k)!} \sum_{\mathcal{S} \subset \{1,\ldots,n+k\}, |\mathcal{S}|=k} L_{\mathcal{S}}(\lambda) \leq \alpha.$$

The proof is then completed by the law of iterated expectation. $\square$

