# OpenReview forum: "Conformal Risk Control"
_ICLR.cc/2024/Conference — ICLR 2024 spotlight_

### Official Review · Reviewer_GsY4 · 2023-10-26

**Soundness:** 4 excellent
**Presentation:** 4 excellent
**Contribution:** 4 excellent
**Rating:** 8
**Confidence:** 4

**Summary:**

This paper generalizes the tools of the Conformal Prediction framework to achieve prediction sets satisfying general criteria, including the traditional coverage criteria. They provide an adaptation to Split Conformal Prediction that achieves this new "risk" metric. They demonstrate empirically that such a method indeed holds similar risk guarantees to the known coverage guarantee over a host of real datasets.

**Strengths:**

1. The intuition of the paper is strong. Developing a method to generalize Conformal Prediction to different loss functions is a clear improvement on existing literature, and the contribution is clear.
2. The experiments are very well done. The experiments over several loss functions are clear and convincing.
3. The paper's motivation is clear, and the impact of such work is immediately evident.
4. The extensions are strong and of good breadth. This framework is evidently useful for providing guarantees under different risk settings, not just the initial canonical setting of risk presented first.

Overall, I vote for acceptance of this paper. I have several questions on computability and the writing, but these can be fixed and explained for the camera-ready version.

**Weaknesses:**

1. Some motivating examples of cases where such conformal risk control would be desirable over the traditional coverage guarantee. One can think of cases where F1-score is more appropriate to measure than coverage. It would be great to mention these examples in the introduction so that the reader understands more the motivation of this generalization.
2. Why is a bounded loss function reasonable? What examples of loss functions violate this or satisfy this? Obviously, a coverage guarantee satisfies this, but what other metrics satisfy this?
3. The $\Lambda$ operator is quite confusing. You do not explicitly define it and still say $L_i$ is a map from $\Lambda$ to another set. You should have defined $\Lambda$ before. You also say $\sup \Lambda \in \Lambda$. Is $\Lambda$ the set of possible $\lambda$, so the set of real numbers?
4. In the buildup of section 1.1, I find the intuitive explanation of the method and why it works lacking. I believe a strong reason of why conformal prediction is so popular is its intuitiveness. I find these definitions unintuitive and the provided explanation lacking. I would spend more time describing the intuition of the constants and claims defined.
5. One of my main concerns is the computability of $\hat{\lambda}$. This seems highly nontrivial to calculate. This seems like the most important point of the paper. Could you describe how to compute this? Moreover, does this formalization make it easier to compute than other similar methods doing conformal risk control?
6. Does the guarantee of $\alpha -\frac{2B}{n+1}$ in Section 1.1 match that of conformal prediction? In that case, the known guarantee is $\geq \alpha$, which is loose by $\frac{2}{n+1}$. Does this method recover the tightest guarantees of Split Conformal Predictions when the loss function is set to be miscoverage?
7. In the related works section, I find the differences between existing and current work confusing. You mention the guarantee in this paper is in expectation. Why is this better than existing works? Do existing works do worst-case analysis or high probability analysis? The technical similarity between different methods is not a sufficient difference in my opinion. In the second paragraph, I miss what the existing conformal risk works do. Is it possible that existing conformal risk works do not directly generalize conformal predictions or are often less tight? I believe more details on the differences between existing conformal risk papers and this paper are needed.
8. I find the theoretical section again void of any intuition presented. I would highly encourage the authors to provide more intuition on the proof as to the meaning of each step in section 2.1. I found it very difficult to parse. Moreover, the notation in the proof of Theorem 1 is difficult to read. I believe a little cleaning up of this proof would be helpful. An example is that using $\hat{\lambda}'$ and $\hat{\lambda}$ are poor notation choices and create a lack of clarity.
9. The assumption in Theorem 2, i.e. $\mathbb{P}(J(L_i, \lambda) > 0) = 0$ is provided without sufficient explanation. What set of loss functions satisfies such a definition and why? Which loss functions do not satisfy this assumption? Providing this analysis will help give intuition as to what Theorem 2 shows and when it works.

**Questions:**

Please see above.

---

> ### Author Response · Authors · 2023-11-20
>
> We thank the reviewer for their careful review and thoughtful comments. We provide answers to specific questions and remarks (quoted) below.
>
> > Some motivating examples of cases where such conformal risk control would be desirable over the traditional coverage guarantee. One can think of cases where F1-score is more appropriate to measure than coverage. It would be great to mention these examples in the introduction so that the reader understands more the motivation of this generalization.
>
> Great idea; thanks for the suggestion! Another reviewer agreed here, and we have bolstered the motivating example to the introduction.
>
> > Why is a bounded loss function reasonable? What examples of loss functions violate this or satisfy this? Obviously, a coverage guarantee satisfies this, but what other metrics satisfy this?
>
> In our Examples section, we give several instances of such loss functions, including the best-in-set F1 score (lower-bounded by 0, so that 1 - F1 is lower-bounded by 1), the false negative rate (upper bounded by 1), graph distance (upper bounded by the depth of the graph), as well as any other clipped loss, such as Tukey's biweight loss.
>
> Furthermore, any unbounded loss can be transformed to a bounded loss. For example, any unbounded loss on the positive reals can be transformed by taking the inverse tangent. As long as the transformation is monotone, a loss of $\alpha$ on the original loss corresponds exactly to a loss of $\alpha’$ on the new loss; thus, controlling the transformed risk may be enough in practice.
>
> We added text to the paper to make this clear; see page 2, footnote 1.
>
> > The  operator is quite confusing.
>
> We introduced $\Lambda$ in a confusing way. $\Lambda$ is just the space of all inputs (‘thresholds’) to the function $L(\lambda)$. It is usually a subset of the real line, but can also be a discretized grid (for example, np.linspace(0, 1, 100)). We have clarified this in the paper.
>
> > In the buildup of section 1.1, I find the intuitive explanation of the method and why it works lacking. I believe a strong reason of why conformal prediction is so popular is its intuitiveness. I find these definitions unintuitive and the provided explanation lacking. I would spend more time describing the intuition of the constants and claims defined. Provide more intuition on the proof as to the meaning of each step in section 2.1. I found it very difficult to parse.
>
> Thanks for another good suggestion. We have added more intuition to the explanation of the method in Section 1.1, which hopefully will help with the proof intuition as well. We would appreciate any further guidance from the reviewer as to what was unintuitive. (It is hard to add much more content to the proof due to the page constraints.)
>
> > One of my main concerns is the computability of $\hat{\lambda}$.
>
> It is computationally trivial to compute $\hat{\lambda}$ (in our new experiments, it takes a fraction of a second). Since the loss is monotonic, it is simple to find  by binary search to arbitrary precision. Usually this requires fewer than ten evaluations of the empirical risk.
>
> We added an explanation of this fact in text (see page 2).
>
> > In the related works section, I find the differences between existing and current work confusing. You mention the guarantee in this paper is in expectation. Why is this better than existing works? Do existing works do worst-case analysis or high probability analysis? The technical similarity between different methods is not a sufficient difference in my opinion. In the second paragraph, I miss what the existing conformal risk works do. Is it possible that existing conformal risk works do not directly generalize conformal predictions or are often less tight? I believe more details on the differences between existing conformal risk papers and this paper are needed.
>
> Thank you for this important criticism, which we took very seriously. Please see the [main response comment]() for a detailed discussion. See also the related work section and Appendix B of the updated paper.
>
> The practical difference between our approach and the existing ones is massive --- it is **substantially more effective, sample-efficient, simpler, and provides a direct theoretical contribution to conformal prediction** as opposed to the existing risk-control algorithms. As the reviewer mentioned, the other approaches are high-probability bounds that do not generalize or build on the standard theory of conformal prediction.

---

> > ### Comment · Reviewer_GsY4 · 2023-11-21
> > **Response to Authors**
> >
> > I'd like to thank the authors for their response. I'd like to keep my vote to accept this paper.

---

### Official Review · Reviewer_fQpV · 2023-11-01

**Soundness:** 3 good
**Presentation:** 3 good
**Contribution:** 3 good
**Rating:** 6
**Confidence:** 3

**Summary:**

This paper extends conformal prediction to control the expected value of any monotone loss function. This type of guarantee called conformal risk control can be seen as a generalization of CP and includes CP as a particular case. Authors prove that for bounded loss, the conformal risk procedure provides a bound for any given $\alpha$ and this bound is tight up to a $\mathcal{O}(1/n)$ factor. The paper also explores different settings/problems such as distribution shift, quantile risk control, adversarial risk control, and expectations of U-statistics. Finally, experiments highlight the soundness of their proposals.

**Strengths:**

This is a clever and promising idea to generalize conformal prediction guarantees.

This problem is of particular interest to the conformal prediction community.

The paper is very well written and easy to follow. This is a real pleasure to read it.

**Weaknesses:**

It seems to me that the paper is more a collection of lots of theoretical results (c.f. Section 4. "Extensions": distribution shift, quantile risk, adversarial risk..)  than a real important contribution to a particular problem. This somewhat blurs the main contribution of the paper, which might have benefited from presenting only the risk control in a standard setting, but with a much more in-depth theoretical analysis.

The experimental results are not analyzed at all. In addition, the extensions in section 4 have not been the subject of any experiments.

No limits are indicated in the conclusion. In my opinion, comments on the limitations of the proposed approach are lacking.

Minor:

The proof of Theorem 1 should be in the appendix with all the other proofs.

"In this case, conformal risk control finds...the expected value is..." , hat on $\lambda$ is missing in the right part of the equality.

Typos: Examining (??),

**Questions:**

Under the distribution shift setting, the weights need to be estimated. How this can be done? and what is the impact on the control of the risk when $w$ is replaced by an estimate?

Is there a link between quantile risk control and training-conditional guarantee in standard conformal prediction?

Can you give a real situation where we want to control risks defined by adversarial perturbations?

The assumption that $P(J(L_i, \lambda) > 0) = 0$ appears to be the CP equivalent of the continuous score assumption. Is this true?

It is said in the paper that in "monotonizing" a loss will only be powerful if the loss is near-monotone. Although this seems intuitive, do you have any numerical evidence for this? (or/and a basic example).

---

> ### Author Response · Authors · 2023-11-20
>
> We thank the reviewer for their careful review and thoughtful comments. We provide answers to specific questions and remarks (quoted) below.
>
> > ...the paper is more a collection of lots of theoretical results (c.f. Section 4. "Extensions": distribution shift, quantile risk, adversarial risk..) than a real important contribution to a particular problem. This somewhat blurs the main contribution of the paper...
>
> **Theorems 1 and 2 are our main contributions, and we believe they are substantial contributions to the theory of conformal prediction.** The main observation is that conformal-type proof strategies leveraging exchangeability can be extended to a broader mathematical setting; specifically, they can be used for any bounded, monotone loss function, not just coverage.
>
> Section 4 is a bit more scattered, but shows the generality the approach by demonstrating its application to a few popular settings elsewhere in the conformal literature.
>
> We don’t feel it is fair to say that our paper is not an important contribution to a particular problem. Theorems 1 and 2 are fundamental contributions; the rest are extensions. If the reviewer feels we have muddied our contribution by including Sec. 4, we are happy to discuss moving it to the appendix.
>
> > The experimental results are not analyzed at all. In addition, the extensions in section 4 have not been the subject of any experiments.
>
> We have added analyses of our experimental results, and an example of covariate shift to Appendix D.
>
> > No limits are indicated in the conclusion. In my opinion, comments on the limitations of the proposed approach are lacking.
>
> We have added a new limitations section, as mentioned in the [main response comment](https://openreview.net/forum?id=33XGfHLtZg&noteId=Y1zeKwutId). It is copied below for convenience.
>
> > Under the distribution shift setting, the weights need to be estimated. How can this be done?
>
> We now include an experiment that estimates the covariate shift weights in Appendix D. It is done by training a binary probabilistic classifier to distinguish the training (Y=0) and test data points (Y=1). The LR estimate is taken to be $f(x)/1-f(x)$. This approach tends to work fairly well in practice (see our new experiment and those in [3]), but further theory on this topic would be very interesting.
>
> [3] Tibshirani, R. J., Foygel Barber, R., Candes, E., & Ramdas, A. (2019). Conformal prediction under covariate shift. Advances in neural information processing systems, 32.
>
> > Is there a link between quantile risk control and training-conditional guarantee in standard conformal prediction?
>
> There is no link between these topics. This is because expectations and quantiles are fundamentally different objects and cannot be related without distributional assumptions.
>
> Training conditional guarantees, give bounds on the conditional distribution of $\mathbb{E}[L_{n+1}(\hat{\lambda}) \mid L_1, \ldots, L_n]$.  Quantile risk control gives a guarantee of the form $\mathrm{Quantile}(L_{n+1}(\hat{\lambda})) \leq \alpha$. Relating these two guarantees is not generally possible.
>
> > Can you give a real situation where we want to control risks defined by adversarial perturbations?
>
> An example could be when a machine learning system is serving exchangeable user queries, but, during deployment, adversaries may have injected bounded perturbations to the test queries. A real such situation might be an image recognition system guiding a car, with a threat of adversarially placed markings on road signs. The adversarially-robust version of conformal risk control maintains exchangeability of the worst case risk in this class of perturbations.
>
> > The assumption that P(J(Li, ) > 0) = 0 appears to be the CP equivalent of continuous score assumption. Is this true?
>
> Yes, these assumptions are analogous. We previously noted this in Appendix A, and edited the main text to include this as well.
>
> > It is said in the paper that in "monotonizing" a loss will only be powerful if the loss is near-monotone. Although this seems intuitive, do you have any numerical evidence for this? (or/and a basic example).
>
> We are happy to present a basic example. Consider the example loss function where $L(\lambda) = 0$ if $\lambda > 1 - \epsilon$, $L(\lambda) = B$ if $\lambda \in [a_i, 1  - \epsilon]$, and $L(\lambda) = 0$ if $\lambda < a_i$. Then we have that $\hat{\lambda} = 1 - \epsilon$, regardless of the distribution of $a_i$. The intuition here is that if there are big, non-monotone blips in the loss for certain $\lambda$, the $\sup_{\lambda' \geq \lambda} L(\lambda')$ version of the loss will be driven high early on, and can remain very conservative relative to the non-monotone $L(\lambda)$ at lower values of $\lambda$.

---

> > ### Comment · Reviewer_fQpV · 2023-11-21
> >
> > I would like to thank the authors for their detailed response. I would also like to stress that I found the main contribution (Theorem 1 and 2) of the article very interesting and useful for the CP community.
> >
> > I maintain my score in favor of accepting the article.

---

### Official Review · Reviewer_pT4W · 2023-11-01

**Soundness:** 3 good
**Presentation:** 3 good
**Contribution:** 3 good
**Rating:** 8
**Confidence:** 3

**Summary:**

The paper offers a generalization of split conformal prediction, from its standard coverage-type guarantees to producing (threshold-based) set-valued predictions which bound, in expectation, arbitrary monotonic (in the threshold selected) loss functions (i.e. risks). The guarantees are very similar to conformal prediction (up to knowing an upper bound on the risk function), and the method flexibly generalizes some other important extensions of conformal prediction, such as e.g. the distribution shift conformal modeling of Tibshirani et al.

**Strengths:**

The paper considers a productive generalization of conformal prediction, and easily extends the same proof technique as in vanilla conformal to handle relevant monotonic risks that are not coverage indicators. This leads to a host of both worked-out, and potential, applications to provably bounding useful risks (such as e.g. the running example of FNR) in various settings.

The useful extensions, to quantile control, multiple risks, and covariate shift among others, are also quite easy to derive from this generalized framework.

The experiments are (for the most part) cleanly formulated and executed, confirming that relevant risks are easily bounded in practice as predicted from the theory.

The paper is overall pleasantly written, and surveys all its contributions in a transparent manner.

As a result, I believe this paper would be a nice contribution to the conference, for both its theoretical soundness and simplicity as well as for its potential to be used in applications as diverse as split conformal prediction itself.

**Weaknesses:**

1. A relatively significant point is that, while theoretical bounds here are given in expectation, as compared to existing PAC-type guarantees on general risks developed in other recent related papers, but in practice it would still make a lot of sense to establish good empirical performance of conformal risk prediction relative to those --- and in particular, to Learn-then-Test, which appears both very tractable just as the proposed framework here, but also handles arbitrary losses, not just ones that are monotonic, and thus seems to offer better flexibility in practical settings (without having to resort to discussions of near-monotonicity like the present manuscript does). Since the displayed metrics about the conformal risk control method are agnostic to whether in-expectation or PAC-style guarantees are given, it should be easy to make this comparison quite direct, from a practical standpoint.

2. As a very minor point, the experiment on F1 score control in open-domain question answering, in contrast to other experiments in this paper, seems less useful and set up in a somewhat stylized way that doesn't seem to aim to capture any of the linguistic difficulties in tackling open domain question-answering --- which makes the plotted performance graph seem not very interesting. The F1 score is computed over a rather simplistic bag-of-tokens similarity between predictions and answers, and is not justified by the authors as the risk measure of choice here; moreover, a shift to alpha = 0.3 happens with the scant explanation that this is the best empirical choice such that almost all answers are typically correct. Recognizing that this is just a toy example to support the validity of the proposed method, I would still have liked to see a somewhat deeper discussion of what monotonic risks one could/should use when dealing with natural language; this would be very useful in the context of recent developments in LLMs.

**Questions:**

As stated above, I would like to see an experimental comparison to other methods offering guarantees on generalized risks; in particular Learn-then-Test.

---

> ### Author Response · Authors · 2023-11-20
>
> We thank the reviewer for their careful review and thoughtful comments. We provide answers to specific questions and remarks (quoted) below.
>
> > ...establish good empirical performance of conformal risk prediction relative to Learn-then-Test, which appears both very tractable just as the proposed framework here, but also handles arbitrary losses... it should be easy to make this comparison quite direct, from a practical standpoint.
>
> Thank you for your suggestion; please cross-reference the [main response comment](https://openreview.net/forum?id=33XGfHLtZg&noteId=Y1zeKwutId). We took this comment quite seriously and added a new subsection to the paper on this comparison, as well as a new experiment showing substantial practical advantages to our approach; see Appendix B. The practical gains are _massive_ compared to RCPS/LTT.
>
> > As a very minor point, the experiment on F1 score control in open-domain question answering, in contrast to other experiments in this paper, seems less useful and set up in a somewhat stylized way that doesn't seem to aim to capture any of the linguistic difficulties in tackling open domain question-answering --- which makes the plotted performance graph seem not very interesting...
>
> We tend to agree that the F1 score that we compute for open-domain QA is simplistic, though it is a common metric in the QA literature (see, e.g., Table 3.7 in the GPT-3 paper, and the definition of F1 evaluation in the popular SQuAD dataset); which is the reason for why we use it. That said, we note that many other metrics relevant to natural language are possible to use instead of F1 similarly. These include things like BERTScore, or the output of a Bradley-Terry style reward model trained from human feedback, as is commonly used in RLHF. Unfortunately, translating from such real-valued scores to what constitutes a desirable target risk score is less intuitive than things like false negative rate, however, and typically requires domain expertise getting a feel for the perceived "quality". Alternatively, we can also directly use human ratings based on Likert scales, though these are more expensive to annotate on a calibration set.

---

> > ### Comment · Reviewer_pT4W · 2023-11-21
> > **Response to Authors**
> >
> > Thank you to the authors for their effort in addressing all reviewers in this thread, including my concerns. I appreciated both the clarifying and the substantive changes to the manuscript, which are quite extensive and will no doubt be useful to future readers. Importantly, the main concern of lacking comparison to LTT has been addressed; having further experiments on that front would be even better, but the added one does convey the point that LTT may give much less tight practical performance with respect to the expected coverage metric. (As a side note here, of course I do concur with the conceptual points relating to the fact that the PAC-style and the expectation guarantees are mathematically different objects, and that the present paper is a much more direct extension to the core conformal theory, but to me the main differentiator between the methods is precisely the practical performance for various monotonic risks --- especially since from a practitioner's perspective, both methods provide guarantees that have very similar verbal descriptions.)
> >
> > Given the nice responses to the raised issues, I continue to support the present paper towards acceptance, and raise my score to 8.

---

### Official Review · Reviewer_nYbV · 2023-11-03

**Soundness:** 4 excellent
**Presentation:** 4 excellent
**Contribution:** 4 excellent
**Rating:** 8
**Confidence:** 4

**Summary:**

This work generalizes the conformal prediction guarantees for coverage to controlling the risk of general monotone (or near monotone) losses, validating the proposed method on various real-life examples.

**Strengths:**

I think this is an exciting work with many strengths
- Method: generalize the CP method to arbitrary monotone loss functions, with extensions & modifications in distributional shift, controlling risk of multiple tasks, adversarial risks, etc.
- Theory: provide the guarantee for the proposed method, and demonstrating the need for monotone functions (Proposition 2) and how to monotonie loss functions (Corollary 1) for the same guarantee
- Experiment: provide extensive and useful illustration of the method in various tasks.

**Weaknesses:**

I think the only weaknesses lie in the experimental comparison.
1. In all examples presented in section 3, the authors only show the satisfactory performance of the proposed method. No comparison against other baselines is provided. If this is due to a lack of existing methods for similar tasks, it would be helpful to highlight this, explain why this is the case, and include doing so as future directions.
2. For tasks mentioned in the extensions, it would be helpful to also illustrate the effectiveness of the proposed method on a subset of them.

**Questions:**

What are limitations and future directions? It would be helpful to discuss these in the conclusion as well.

---

> ### Author Response · Authors · 2023-11-20
>
> We thank the reviewer for their careful review and thoughtful comments. We provide answers to specific questions and remarks (quoted) below.
>
> > In all examples presented in section 3, the authors only show the satisfactory performance of the proposed method. No comparison against other baselines is provided. If this is due to a lack of existing methods for similar tasks, it would be helpful to highlight this, explain why this is the case, and include doing so as future directions.
>
> Thank you for the good point; we have used this as an opportunity to implement a comparison to LTT/RCPS [1,2], which are high-probability versions of our technique. See the [main response comment](https://openreview.net/forum?id=33XGfHLtZg&noteId=Y1zeKwutId) for details.
>
> Initially, we had not included them because they will clearly be much more conservative than conformal risk control. However, we agree that it is good to understand the gains of our new procedure. We have made major efforts towards understanding this in the revision.
>
> It is worth noting that no other existing methods provide guarantees of risk control in expectation. The main purpose of Section 3 is to demonstrate the flexibility and empirical effectiveness (i.e., validity and tightness) of the proposed algorithm as a tool for risk control, as there are no competing strategies. We have clarified this in the related work.
>
> > For tasks mentioned in the extensions, it would be helpful to also illustrate the effectiveness of the proposed method on a subset of them.
>
> Agreed; to deal with the space constraints, we have added an experiment on covariate shift to Appendix D. It includes estimated covariate weights.
>
> > What are limitations and future directions? It would be helpful to discuss these in the conclusion as well.
>
> We have added a section on the limitations of our work in the discussion; it is copied below for convenience.
>
> This generalization of conformal prediction broadens its scope to new applications, as shown in Section 3. Still, two primary limitations of our technique remain: firstly, the requirement of a monotone loss is difficult to lift. Secondly, extensions to non-exchangeable data require knowledge about the form of the shift. This issue affects most statistical methods, including standard conformal prediction, and ours is no different in this regard. Finally, the mathematical tools developed in Sections 2 and 3 may be of independent technical interest, as they provide a new, and more general, language for studying conformal prediction, along with new results about its validity.
>
> [1] Angelopoulos, A. N., Bates, S., Candès, E. J., Jordan, M. I., & Lei, L. (2021). Learn then test: Calibrating predictive algorithms to achieve risk control. arXiv preprint arXiv:2110.01052.
>
> [2] Bates, S., Angelopoulos, A., Lei, L., Malik, J., & Jordan, M. (2021). Distribution-free, risk-controlling prediction sets. Journal of the ACM (JACM), 68(6), 1-34.
>
> [3] Tibshirani, R. J., Foygel Barber, R., Candes, E., & Ramdas, A. (2019). Conformal prediction under covariate shift. Advances in neural information processing systems, 32.

---

### Official Review · Reviewer_TBcZ · 2023-11-07

**Soundness:** 3 good
**Presentation:** 3 good
**Contribution:** 2 fair
**Rating:** 6
**Confidence:** 4

**Summary:**

The paper extends conformal prediction to conformal risk control for any monotone loss function, which can be applied to different tasks more generally, such as token-level F1 scores. They also consider distribution shifts quantified by TV distribution distance and provide evaluations to validate the results.

**Strengths:**

1. The research question is important for many perspectives, such as trustworthiness ML.
2. The paper is comprehensive and includes multiple perspectives besides the main result, including distribution shifts and discussions of different types of losses.

**Weaknesses:**

1. I suggest adding detailed discussions of differences to related work [1,2].

[1] Learn then test: Calibrating predictive algorithms to achieve risk control. arXiv preprint arXiv:2110.01052, 2021

[2]  Distribution-free, risk-controlling prediction sets. Journal of the ACM (JACM), 68(6):1–34, 2021

2. The preview (sec 1.1) is clear, but theorem 1 is not well presented. $\lambda_{\text{max}}$ and $L_i$ are not defined or referred to in the context.

3. It is interesting to discuss whether we can have inverse relations of theorem 1? Specifically, given a parameter $\lambda$, can we inversely compute $\alpha$ to control the risk of $\lambda$?

I'll raise the scores once the problems are well discussed.

**Questions:**

Please refer to the weakness part.

---

> ### Author Response · Authors · 2023-11-20
>
> We thank the reviewer for their careful review and thoughtful comments. We provide answers to specific questions and remarks (quoted) below.
>
> > I suggest adding detailed discussions of differences to related work [1,2].
>
> Thank you for your suggestion; please cross-reference the [main response comment](https://openreview.net/forum?id=33XGfHLtZg&noteId=Y1zeKwutId). We took this comment quite seriously and added a new subsection to the paper on this comparison, as well as a new experiment showing the advantages of our approach. The edits are extensive and we hope they satisfy your concern.
>
> > Theorem 1 is not well presented.
>
> We appreciate the comment. We have defined $\lambda_{\mathrm{max}}$ and $L_i$ in the theorem for the purpose of clarity and rigor, so a reader can understand the result without having to comb through surrounding text. Hopefully this addresses your concern.
>
> > It is interesting to discuss whether we can have inverse relations of Theorem 1?
>
> While we cannot bound $\mathbb{E}[L_{n+1}(\lambda)]$ directly, we can give a probabilistic bound. For example, given a fixed $\lambda$, we can show that $\mathbb{P}(L_{n+1}(\lambda) \leq \mathrm{Quantile}(1 - \delta, \{L_1(\lambda), \ldots, L_{n}(\lambda), B\}) \geq 1 - \delta$. The proof is analogous to that of standard conformal prediction.
>
> Estimating risk functions like $E[L_{n+1}(\lambda)]$ is a well-studied problem in the area of empirical process theory. It is well-known to be an intrinsically $1/\sqrt{n}$ problem (by the Central Limit Theorem, for example), and can be addressed by various concentration inequalities, but not by any conformal method, because they do not estimate population parameters of the data distribution.

---

> ### Author Response · Authors · 2023-11-22
> **Discussion?**
>
> We are hoping to have a discussion today before the deadline passes. Have we addressed your comments?
>
> Thank you again for the review.

---

> ### Comment · Reviewer_TBcZ · 2023-11-22
> **Reviewer response**
>
> Thank you for the clarifications. My concerns are addressed. I raised my score to 6.

---

### Official Review · Reviewer_tJUM · 2023-11-08

**Soundness:** 3 good
**Presentation:** 3 good
**Contribution:** 3 good
**Rating:** 6
**Confidence:** 3

**Summary:**

The paper extends CP coverage guarantees to risk upper bounds. Under the assumption that the risk function is monotone, the authors derive a finite-sample and distribution-free bound on the risk expectation. Extensions and applications of the idea are also provided.

**Strengths:**

Extending the finite-sample and distribution-free coverage guarantees of CP to more general risk minimization problems may have a great practical impact. The experiments section contains a nice series of practical applications. I appreciate the authors included a full proof in the main text.

**Weaknesses:**

The relevance and novelty of the theoretical parts may be stated better in the introduction. The authors should
- clarify why obtaining expectation-based bounds is more challenging than applying existing risk-control algorithms, e.g. the
hypothesis testing strategy of [1], and
- specify what is different and what is taken from other works, e.g. some of the definitions are similar to [1], where the setup is slightly different.

Under the monotonicity assumption, Theorem 1 seems to be a straightforward reformulation of the standard validity proof for CP prediction intervals. Indeed, Section 4.2 outlines a much simpler formulation of the proposed algorithm. The authors may consider moving and discussing that section before Theorem 1 and adding a practical example where Theorem 1 is needed.

**Questions:**

- The algorithm's main parameter, $\lambda$, is introduced in an example. It would be better to have a formal definition of it.
- A motivation is mentioned at the end of page 2: "it is generally impossible to recast risk control as coverage control". The statement is not proven or supported by examples.
- Why does $R_n$ carrry an index? Are $R_1, ..., R_{n-1}$ used anywehere? This seems similar to [1], where the index refers to a specific value of $\lambda$.
- Can $R(\lambda)$ be interpreted as a parameterized conformity score and $\lambda$ as the quantile of the corresponding empirical distribution?
- [1] addresses the case of non-monotonic risks. How is this compatible with Proposition 2?
- Does the method apply to the outputs of a regression model?
- Are the extensions of Section 4 straightforward consequences of Theorem 1 and existing CP theory?

[1]
"Learn then Test: Calibrating predictive algorithms to achieve risk control".

---

> ### Author Response · Authors · 2023-11-20
>
> We thank the reviewer for their careful review and thoughtful comments. We provide answers to specific questions and remarks (quoted) below.
>
> > clarify why obtaining expectation-based bounds is more challenging than applying risk-control algorithms, e.g. [1].
>
> Please cross-reference the [main response comment](https://openreview.net/forum?id=33XGfHLtZg&noteId=Y1zeKwutId) for a detailed comparison of our approach against [1,2].
>
> The expectation bounds in this paper are a novel and direct contribution to the theory of conformal prediction. From that perspective, we believe they are valuable. The high-probability bounds of [1]  rely on extensive existing theory for deriving concentration bounds (e.g. the Hoeffding bound). Our manuscript *contributes more directly to the theoretical foundations of conformal prediction* than [1,2], which are applications of empirical process theory/multiple testing and not standard conformal prediction.
>
> > Compare to other works; clarify similarity of setup to [1]
>
> See the [main response comment](https://openreview.net/forum?id=33XGfHLtZg&noteId=Y1zeKwutId) and edits to related work Basically, the setups are almost the same, but the algorithms and theory are entirely different and the proofs are mathematically unrelated.
>
> > Theorem 1 seems to be a straightforward reformulation of the standard validity proof for CP prediction intervals.
>
> **Theorem 1 is not a reformulation of the validity theorem of conformal prediction, and the extension of conformal prediction to this setting is mathematically nontrivial.** Our algorithm handles a strictly larger class of problems. When the loss function is binary, it reduces exactly to conformal prediction; see Appendix A for a detailed explanation.
>
> In the writing of the proof of Theorem 1, highlighted the similarities to the original conformal validity proof for pedagogical reasons; however, **this proof is not the same as the standard proof of conformal prediction’s validity involving score functions**.  For non-binary loss functions, **no score function exists**; hence the standard proof is inapplicable.
>
> > “Impossible to recast risk control as coverage control.” The statement is not proven or supported by examples.
>
> Thank you for raising the point. **All examples in Sec. 3 are impossible to equivalently cast in terms of coverage control.**
>
> It is not hard to see that risk control is not the same as coverage control. Comparing the form of equations (2) and (1),  if you take $\ell$ to be any non-indicator function, (2) cannot be rewritten in the same form as (1). This is because (1) measures the probability of a miscoverage event, and when $\ell$ is not an indicator, there is no such event to measure.
>
> However, (2) recovers (1) in the case of a binary loss (see intro+Appendix A).
>
> > Why does $\hat{R}_n$ carry an index?
>
> $\hat{R}_n = \frac{1}{n}(L_1 + \ldots + L_n)$ is a standard notation from empirical process theory for an empirical risk on $n$ points ($L_1, \ldots, L_n$).
>
> > Can $R(\lambda)$ be interpreted as a parameterized conformity score and $\hat{\lambda}$ as the quantile of the corresponding distribution?
>
> Good question; no, it cannot. The conformal score only exists for binary losses.
>
> If $L_i$  is binary, the conformity score can be equivalently expressed as a discontinuity in the indicator function, $s_i = \inf \{ \lambda : L_i(\lambda)  = 0 \}$. In that case, $\hat{\lambda}$ is the standard conformal quantile, as in Appendix A. In this setup, $R(\lambda)$ can be interpreted as the empirical miscoverage when sets are constructed using parameter $\lambda$.
>
> However, when $\ell$ is not an indicator function, the conformity score can no longer be defined at all, and this correspondence ceases to exist. To see this, suppose there is a conformity score; then the corresponding loss function has at most one turning point at the threshold of the conformity score. Clearly, this does not hold for general monotone loss functions.
>
> > [1] addresses the case of non-monotonic risks. How is this compatible with Proposition 2?
>
> Proposition 2 does not apply to the algorithm in [1], which is different (much more conservative and complicated) and applies to non-monotonic risks. See the [main response comment](https://openreview.net/forum?id=33XGfHLtZg&noteId=Y1zeKwutId) for a full comparison of these algorithms.
>
> > Does the method apply to the outputs of a regression model?
>
> Certainly! See Appendix D.
>
> > Are the extensions of Section 4 straightforward consequences of Theorem 1 and existing CP theory?
>
> None of the extensions in Section 4 are literally corollaries of existing results + Theorem 1. Some are straightforward to an expert, and some are not.
>
> Most of the results in Section 4 follow by arguing that some modified loss function satisfies the conditions required for Theorem 1 to hold. Propositions 4 and 8, however, use different proof techniques. Of all the propositions, we believe that 4 and 8 are the least straightforward.

---

> ### Author Response · Authors · 2023-11-22
> **Discussion?**
>
> We are hoping to have a discussion today before the deadline passes. Have we addressed your comments?
>
> Thank you again for the review.

---

> > ### Comment · Reviewer_tJUM · 2023-11-23
> > **Thank you for your answers**
> >
> > Thank you for the detailed answers and explanations. I raised my score to 6. What do you mean by "the corresponding loss function has at most one turning point at the threshold of the conformity score"?

---

### Author Response · Authors · 2023-11-20
**Main Response Comment**

We were delighted to see six detailed reviews of our paper. Thank you to all the reviewers for taking the time to read and comment upon our work. We have made major changes to the paper, and uploaded a revised draft with changes in red.

There was one main concern raised by all reviewers: a comparison is needed to the existing literature on risk control, specifically the LTT/RCPS procedures [1,2]. We have taken this feedback to heart, and incorporated the following changes:

1. We added a new subsection titled **Comparison with RCPS/LTT** containing the following content:

      a.**The LTT/RCPS procedures are substantially less sample-efficient than conformal risk control.** On the scale of the risk, LTT/RCPS converge to $\alpha$ at a $\frac{1}{\sqrt{n}}$ rate, while conformal risk control converges at a $\frac{1}{n}$ rate.

      b. LTT/RCPS are high probability bounds of the form $\mathbb{P}( \mathbb{E}[\ell(Y, \mathcal{C}\_\lambda(X)) \mid \text{calibration data}] \leq \alpha)\geq 1-\delta$. **Conformal risk control provides a simpler bound, of the form $\mathbb{E}[\ell(Y, \mathcal{C}\_\lambda(X))] \leq \alpha$.** The simpler bound tends to be **easier to use** and explain to non-experts, since it doesn’t require the selection of $\delta$. **LTT/RCPS are substantially more conservative and complicated.**

      c. The LTT procedure allows for the high-probability control of non-monotone risks, while conformal risk control only applies to monotone ones.

      d. The theory of conformal risk control is a **basic advancement to the core theory of conformal prediction** and exchangeability-based arguments, extending them to a broader mathematical setting. Meanwhile, LTT/RCPS use well-established concentration bounds, which are unrelated to conformal prediction.

2.  The paper now includes a careful evaluation against these procedures on the tumor segmentation dataset. We compare against two versions of LTT/RCPS: **Baseline 1** has $\delta=0.5$, and **Baseline 2** integrates the tail bound of LTT/RCPS to achieve a bound in expectation. The former strategy is not statistically valid for the goal of expectation control, and the latter strategy is essentially only possible using fixed-width bounds such as Hoeffding’s inequality, and is described in Appendix B. We use the same risk level $\alpha=0.1$ for all procedures and make plots with $n=\{25, 50, 100, 200, \ldots, 300\}$. The takeaways are as follows:
      a. **The statistical efficiency gains of conformal risk control are massive.** On the scale of the risk gap, $|\mathbb{E}[R(\hat{\lambda})] - \alpha|$, conformal risk control is 50% closer to the desired risk than Baseline 1, and orders of magnitude closer than Baseline 2.

      b. **The computational efficiency of conformal risk control is substantially better than all baselines.** For example, it is 14.3x faster than Baseline 1 and 19% faster than Baseline 2. It is also substantially easier to implement, only requiring 5 lines of code, while Baselines 1 requires dozens. Baseline 2 is equally easy to implement if Hoeffding’s inequality is used, but much harder otherwise.

We have also added a **new experiment on covariate shift** (see Appendix D) and a **paragraph on the limitations of our work to the discussion section**.

Specifically, the main limitations of our technique are twofold: firstly, the requirement of a monotone loss is difficult to lift. We have attempted to come up with a conformal-type procedure to control non-monotone risks, but were unable to without monotonizing the loss. Secondly, as in standard conformal prediction under distribution shift, our extensions to distribution shift presented here also require knowledge about the form of the shift; specifically, it requires some form of likelihood ratio. Although there are techniques for doing this in the case of covariate shift by training a classifier on unlabeled data (see our covariate shift experiment in Appendix D, or the experiments in [3]), there are many distribution shifts for which it may not be possible to run our methods. This issue affects most statistical methods, and ours is no different in this regard.

We hope this addresses the main concerns of the reviewers. The remainder of the concerns are individualized, and we will respond in separate comments.

[1] Angelopoulos, A. N., Bates, S., Candès, E. J., Jordan, M. I., & Lei, L. (2021). Learn then test: Calibrating predictive algorithms to achieve risk control. arXiv preprint arXiv:2110.01052.

[2] Bates, S., Angelopoulos, A., Lei, L., Malik, J., & Jordan, M. (2021). Distribution-free, risk-controlling prediction sets. Journal of the ACM (JACM), 68(6), 1-34.

[3] Tibshirani, R. J., Foygel Barber, R., Candes, E., & Ramdas, A. (2019). Conformal prediction under covariate shift. Advances in neural information processing systems, 32.

---

### Meta-Review · Area_Chair_5vak · 2023-12-05

**Metareview:**

Conformal prediction is a methodology for associating prediction with some measure of confidence. Given an input observation $x$, It translates as a confidence set on the corresponding output $y$ that has a controlled coverage guarantee that is valid for any data distribution and finite sample size. However, such a control on confidence set is not always enough and one would like, for instance, to control the expected number of miss-classification or other problem related loss. This paper propose such a generalization.

The theoretical results are clearly stated and the proofs are quite transparent.

I also find the benchmark on practical ML problems such as NLP and computer vision quite valuable to demonstrate the usefulness of the framework.

As a minor weakness, I would mention that the results are established in a data splitting settings. Extension such as the full CP, or leveraging stability or cross-validated version would be nice. That being said, I believe that the paper contains enough novelties for a clear accept at ICLR conference.

**Justification For Why Not Higher Score:**

The contributions of this paper are significant as the extension of classical conformal prediction can be quite useful.
I did not recommend higher score because I do believe that confidence sets on the loss $L_{n+1}(\lambda)$ that can be attained are actually more informative and *safer* than a control that guarantee $L_{n+1}(\lambda) \leq \alpha$ in *expectation*, that might not be informative for particular data samples.

That being said, this is a biased point of view and the paper can be considered for an oral presentation as well.

**Justification For Why Not Lower Score:**

The paper is clearly above the acceptance threshold and I can expect it to be quite impactful despite some of the limitations.

I am quite certain about the acceptance decision, which I also believe faithfully represent the reviewers opinion.

---

### Decision · Program_Chairs · 2024-01-16

Accept (spotlight)